



**Large Ozone Intrusions during Sudden Stratospheric Warmings Enhance**
**Ozone Radiative Forcing over South Asia**
Shubhajyoti Roy[1], Satheesh Chandran PR[1], Suvarna Fadnavis[1*], Vijay Sagar[1], Michaela I.
Hegglin[2], Rolf Müller[2]
[1]Centre for Climate Change Research, Indian Institute of Tropical Meteorology, India
[2]Institute of Energy and Climate Systems: Stratosphere (ICE-4), Forschungszentrum,
Jülich, Germany
*Corresponding author email: suvarna@tropmet.res.in





**Abstract**
Tropospheric ozone pollution in South Asia is mainly blamed on anthropogenic
emissions. However, this study highlights the contribution of stratospheric ozone
intrusions into the troposphere associated with sudden stratospheric warming (SSW)
events in enhancing tropospheric ozone over the South Asian region using ERA-5
reanalysis data. We report that specifically split-downward propagating SSWs (dSSWs)
cause enormous ozone enhancement in the upper troposphere and lower stratosphere
(UTLS) over South Asia around the dSSW-onset, with a maximum of ~290% within ±30
days. The ozone intrusions propagate deep into the troposphere, causing near-surface
maximum ozone increase by 43% within ±30 days around the SSW-onset. The ozone
enhancement increases ozone radiative forcing in the troposphere by $0.04\pm0.03$ W.m$^{-2}$
and UTLS by $0.08\pm0.06$ W.m$^{-2}$ over South Asia. Frequent SSW events in a warming
climate will thus likely increase stratospheric ozone intrusions and ozone radiative
forcing over South Asia, potentially exacerbating regional climate warming. The elevated
tropospheric ozone amounts due to stratospheric intrusions are posing threat to humans
and vegetation.
Keywords: Sudden stratospheric warming, stratosphere intrusions, ozone radiative
forcing, South Asian region, Rossby wave breaking.
**1. Introduction**
Tropospheric ozone is a short-lived greenhouse gas that plays a crucial role in
atmospheric chemistry and radiative forcing (Wang et al., 2022). It is also a major air
pollutant that significantly affects human health (Lim et al., 2012; Fleming et al., 2018),
damages vegetation (Fowler et al., 2009; Feng et al., 2021), disrupts ecosystems, and imposes
economic costs (Dewan and Lakhani, 2022). In South Asia, a large amount of tropospheric



ozone is a growing concern because of its ill effects, resulting in rising mortality rates (Silva
et al., 2013; Lin et al., 2018).
The increase in tropospheric ozone levels in South Asia is primarily attributed to
enhanced anthropogenic emissions (Rathore et al., 2023). However, the contribution from the
downward transport of ozone-rich air from the stratosphere is the largest natural source of
tropospheric ozone. Studies have reported that stratospheric influence on the tropospheric
ozone exceeds 50% in the winter season at the extra tropics (Williams et al., 2019). Wang
and Fu (2021) estimate that stratosphere-to-troposphere exchange (STE) contributes
approximately 347±12 Tg year$^{-1}$ to the global tropospheric ozone budget based on both
observations and reanalysis data. CMIP6 models suggest that up to 30% of surface ozone in
the Northern Hemisphere during winter (DJF) is being attributed to stratospheric ozone
intrusions (Li et al., 2024). In the Northwest Pacific, STE increases mid and upper-
tropospheric ozone by about 96% in winter and 40% in summer (Ma et al., 2024). Numerous
observational and reanalysis studies confirm that stratospheric intrusions enhance surface
ozone levels over East Asia and the Tibetan Plateau by ~15 ppb (e.g., Ou-Yang et al., 2022;
Yin et al., 2023). Roy et al. (2023) reported an ozone enhancement of ~40 ppb in the upper
troposphere over the Indian region due to stratospheric intrusions associated with tropical
cyclones.
Sudden stratospheric warming (SSW) events are significant drivers of STE, playing a
key role in atmospheric dynamics and stratospheric ozone intrusions into the troposphere
(e.g., Williams et al., 2024). SSWs are one of the most significant large-scale dynamical
phenomena occurring in the stratosphere during winter ( Butler et al., 2015; de la Cámara et
al., 2018; Baldwin et al., 2021). Enhanced planetary wave activity from the troposphere
disrupts the stratospheric polar vortex, decelerating or even reversing the stratospheric



westerlies, and causing a rapid rise in polar stratospheric temperatures by up to 50 K within
just a few days (Baldwin et al., 2021). SSW events play a crucial role in modulating
tropospheric weather phenomena, such as extreme heat, air pollution, wildfires, wind
extremes, storm clusters, tropical cyclones, and sea ice melt in the northern high latitudes (;
Domeisen and Butler, 2020; Domeisen et al., 2020). The temperature and wind anomalies
associated with SSWs propagate downward into the troposphere over timescales ranging
from weeks to months, impacting surface weather in the Northern Hemisphere for up to 40
days following the event onset (Baldwin and Dunkerton, 2001; Hall et al., 2021). Projection
studies suggest that SSW events will increase by approximately one event per decade by the
end of the 21$^{st}$ century (Charlton-Perez et al., 2008), and high greenhouse gas emission
scenarios show a doubling in SSW frequency (Schimanke et al., 2012). Considering the
frequent occurrences and the potential role of SSWs in STE, it is crucial to investigate SSW's
influence on tropospheric ozone enhancements and the associated radiative effects.

SSW events are classified into two categories, namely displaced or split events, based

on the geometry of the polar vortex (Charlton and Polvani, 2007). In the displaced case, the
vortex is displaced off the pole, while it is split into two baby vortices in the split case.
Further, SSWs are classified as downward propagating and non-downward propagating.
Downward propagating SSWs (dSSWs) show a downward progression of polar cap height
anomalies across vertical levels that reach the surface and exhibit strong surface impacts,
while this is not the case for non-downward propagating SSWs (nSSWs) (Hall et al., 2021).
dSSWs lead to long-lasting tropospheric circulation changes in contrast to nSSWs (;
Karpechko et al., 2017). The dSSWs are often followed by an equatorward shift of the
tropospheric jet stream and storm tracks, as well as surface pressure anomalies that resemble
the negative phase of the Northern Annular Mode (Sigmond et al., 2013; Kidston et al.,
2015). Among dSSWs, studies indicate that the surface effects of split events typically appear





nearly a week earlier than those of displaced SSWs (Mitchell et al., 2013; Hall et al., 2021).
Furthermore, CMIP5 models suggest that split events tend to propagate downward to the
surface more quickly than displacement events (Hall et al., 2021).

SSW events significantly influence STE and impact the tropospheric ozone budget,

particularly in high-latitude regions (Xia et al., 2023; Lu et al., 2023; Williams et al., 2024).
Based on SSW events from 1980–2013 and chemistry-climate model simulations, STE led to
an average 5–10% increase in near-surface ozone over the Arctic (Williams et al., 2024). Xia
et al. (2023) reported an even more pronounced increase of 76% in Arctic surface ozone due
to STE in the 2020/21 SSW event. While most of these studies focus on the polar regions,
some have identified SSW-induced ozone variability in the mid-latitudes as well (Liu et al.,
2009; Lu et al., 2022; Williams et al., 2024). For example, Lu et al. (2022) demonstrated that
meteorological changes associated with SSWs cause poor air quality in the Beijing-Tianjin-
Hebei region. Liu et al. (2009) noted an ozone enhancement of about 186 Tg in the upper
troposphere over East Asia during the 2002–2003 SSW, using MOZART-3 simulations.
However, tropospheric ozone variations during SSW events over South Asia are among the
least studied. Additionally, the broader implications of these events on the ozone radiative
forcing over this region remain largely underexplored. In this study, we investigate the
impact of all the downward-propagating SSW events from 1962 to 2018 on tropospheric
ozone variability over South Asia [20-35°N, 65-90°E] using ERA5 data. The composite is
obtained by averaging data with the onset day as a central date (details in the 'Methods'
section). Here, we report enormous ozone enhancement in the troposphere over South Asia,
leading to an increase in ozone radiative forcing, which further elevates warming in South
Asia. We present a detailed mechanism for the latest 2018 split-dSSW event and a composite
analysis of all split-dSSW events from 1962 to 2018. The net radiative forcing estimation
using the radiative kernel approach is described in the 'Methods' section.



## 2. Methods

### 2.1 ERA 5 Data

We analysed daily data of ozone, zonal and meridional winds, geopotential height, and potential vorticity (PV) from the fifth-generation reanalysis dataset (ERA5) provided by the European Centre for Medium-Range Weather Forecasts (ECMWF) (Hersbach et al., 2020). The ERA5 data, with a horizontal resolution of 0.25°×0.25° and 37 pressure levels ranging from 1000 hPa to 1 hPa, were utilized for this study. Composite analysis was conducted for all variables for a 121-day period centred on the onset of SSW events (60 days before and after the onset) to assess the impact. Daily anomalies in ozone, geopotential height, winds, and PV during the SSW days were calculated by subtracting the corresponding daily mean of all the non-SSW days for ±61 days near the onset of the SSW. The anomalies obtained from climatology (1962-2018) also show features similar to those when one use the mean of all the non-SSW days (see Fig. S1. and Fig. 1a-b). We prefer to use the mean of all the non-SSW days instead of climatology since climatology includes SSW events.

The onset of each SSW event is identified as the day when the zonal mean westerly winds at 10 hPa and 60°N reverse their direction from westerlies to easterlies (Charlton and Polvani 2007). Figure S2. shows the temporal evolutions of the zonal-mean zonal wind at 60° N and 10 hPa for all the split-dSSWs considered for the present study.

### 2.2 Computation of ozone radiative forcing

The radiative forcing (RF) due to ozone is estimated using an ozone radiative kernel method (Skeie et al 2020). The radiative kernel is constructed using the University of Oslo radiative transfer model (Myhre et al., 2011) by perturbing the ozone layer by layer. Temperature, water vapour, and clouds are incorporated into the model from ECMWF's





forecast for the year 2003 and applied as monthly averages. The model calculates radiative
forcing using a broad-band scheme for longwave (Myhre and Stordal, 1997) and DIScrete
Ordinate Radiative Transfer code for shortwave (Stamnes et al., 1988). Previous studies show
that the ozone radiative forcing estimates from the radiative-kernel technique and radiative
transfer model agree within 0.01 W.m$^{-2}$ globally (Iglesias-Suarez et al., 2018). Before the
application of kernel, the ERA5 ozone data is linearly interpolated to the kernel resolution
(~5.6° × 5.6° horizontal with 60 vertical levels). This interpolated ozone field is converted
into Dobson units following Ziemke et al. (2001) and is multiplied with the kernel to estimate
the RF. The tropospheric ozone RF is determined by summing the RF contributions from all
atmospheric layers between the surface and the tropopause (Shell et al., 2008). A similar
approach is applied to estimate radiative forcing for the UTLS region and the total
atmosphere. The tropopause pressure is identified based on the WMO lapse rate tropopause
definition.
**2.3 Dynamical changes in PV, GPH and ozone in the stratosphere during the 2018 event**

The time evolution of the vortex structure depicted by PV at 10 hPa for ±60 days

around the 2018 SSW onset is shown in Fig. S3. Since SSW effects are seen for ±60 days
around the onset (Limpasuvan et al., 2004; Scheffler et al., 2022), we analysed the evolution
of SSW during these days. As the SSW event approaches, the vortex begins to elongate and
become asymmetrical due to the influence of planetary waves propagating upward from the
troposphere, such deformation of vortex is reported in the past (e.g., Baldwin et al., 2021).
Sixty days before onset, a strong, stable polar vortex is evident, but as the event approaches,
planetary wave activity causes elongation and asymmetry (Fig. S3). On the onset day (12
February), the vortex splits into two high-PV lobes over Eurasia and North America (Fig.
S3h). Following the onset, baby vortices exhibit swirling and filamentation, with the Eurasian



lobe drifting westward. Earlier, de la Camara et al. (2018) demonstrated that planetary-scale wave breaking intensifies mixing and facilitates the diffusion of PV from the vortex by elongating and stirring the PV fields. These PV variations align with changes in GPH and ozone fields (e.g., Baldwin et al., 2021), emphasizing stratospheric circulation changes (Fig. S3).

Since major changes in the vortex occur ±6 days around the onset, we show the variations in ozone and GPH during this period (Fig. S4). The GPH anomalies at 10 hPa highlight stratospheric circulation changes, showing a transition from a wave-1 to a wave-2 pattern just before onset. On onset day, strong positive GPH anomalies appear over the Arctic, while negative anomalies correspond to the baby vortices (Fig. S4a-h). This pattern persists for six days and weakens as positive anomalies extend into the United States. Ozone anomalies follow a similar pattern (Fig. S4i-p), with negative values inside the vortex during the pre-onset period due to chemical loss (Manney et al., 2015; Baldwin et al., 2021). After onset, the transport of ozone-rich air leads to positive anomalies at the North Pole. A similar sudden increase in the transport of ozone-rich and high GPH air to the polar region on the onset days is reported by many other studies (Bouillon et al., 2023; Veenus et al., 2023; Shi et al., 2024). Following the onset, positive ozone anomalies spread southward along two branches, one along the northern parts of Africa, Eurasia and the Indian subcontinent and the other over the Atlantic and southern US. This indicates the transport of ozone-rich air towards lower latitudes. Several studies have reported earlier that SSWs cause stratosphere-troposphere coupling in the mid and low latitudes (e.g., Gomez-Escolar et al., 2014; Albers et al., 2016; Williams et al., 2024). The disruption of the polar vortex and the resulting stratospheric conditions redirect planetary waves toward lower latitudes, leading to significant tropospheric circulation changes (Gomez-Escolar et al., 2014).



## 3. Results

### 3.1 Vertical variation of Ozone over the South Asian region during SSW events

We investigated all SSW events from 1962 to 2018 to assess their impact on ozone variability in the upper troposphere over the South Asian region. The categorization of split/displaced and downward (dSSW)/non-downward (nSSW) propagating SSW is as per Hall et al., (2021). Table 1 lists the split-dSSW, displaced-dSSW, and nSSW events considered in this study.

**Table 1**. List of downward propagating split SSW events from 1962 to 2018 considered for the present analysis alongside their onset dates.

| Split dSSW | Onset day | Displaced dSSW | Onset day | non-downward SSW | Onset day |
|---|---|---|---|---|---|
| 1963 | 28 January | 1965 | 16 December | 1966 | 23 February |
| 1968 | 7 January | 1968 | 28 November | 1969 | 13 March |
| 1971 | 20 March | 1980 | 29 February | 1970 | 2 January |
| 1977 | 9 January | 1981 | 4 March | 1971 | 18 January |
| 1979 | 22 February | 1981 | 4 December | 1973 | 31 January |
| 1985 | 1 January | 1984 | 24 February | 1987 | 23 January |
| 1988 | 14 March | 1998 | 15 December | 1987 | 8 December |
| 1999 | 26 February | 2000 | 20 March | 1989 | 21 February |
| 2009 | 24 January | 2001 | 11 February | 2001 | 30 December |
| 2010 | 9 February | 2004 | 5 January | 2003 | 18 January |
| 2013 | 6 January | 2006 | 21 January | 2007 | 24 February |
| 2018 | 12 February | 2008 | 22 February | | |
| | | 2010 | 24 March | | |

We analyzed ozone variations in the UTLS during split-dSSW, displaced-dSSW, and nSSW events. During the SSW event, the disrupted vortex couple with the troposphere, causing shifts in the tropospheric westerly jet and distinctive patterns of anomalous surface temperature and sea-level pressure over a period up to 60 days after SSW onset (e.g., Mitchell et al., 2013; Butler et al., 2017). Figure 1a-d illustrates spatial maps of ozone



anomalies in the UTLS over the South Asian region, averaged over 30 days prior to and after
SSW onset (±30 days) for the aforementioned cases. There is a distinct enhancement in ozone
levels in the UTLS, in the 20°–35°N belt, by 8–16% (40 – 45 ppbv) in the 2018 split-dSSW
event and by 4–10% (35 – 40 ppbv) in the composite of all split-dSSWs compared to the non-
SSW climatology (Fig. 1a-1b). Ozone enhancement in the UTLS is not seen for ±30 days in
the case of displaced-dSSW and nSSW events (see Fig. 1c-d). This highlights the importance
of considering split-dSSW events when assessing the impact of SSWs on ozone variability in
the South Asian region. Ozone enhancement is associated with Rossby wave breaking
(RWB) in the vicinity of the South Asian region which is seen only in the case of split-dSSW
(discussed in section 2.2).

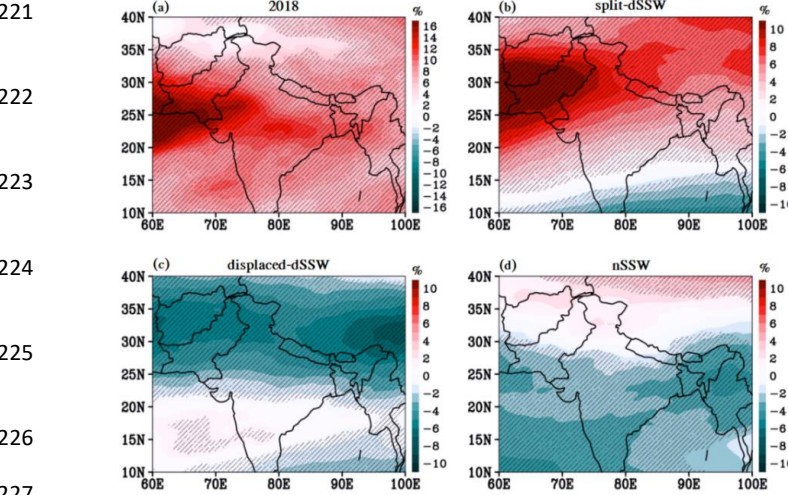

**Figure 1.** Composite map of ERA5 ozone anomalies in the UTLS (250–50 hPa) over the South Asian region, averaged from ± 30 days around the onset for (a) the 2018 split-dSSW event, (b) all split-dSSWs (12 events), (c) all displaced-dSSWs (12 events), and (d) all nSSWs (9 events) as listed in Table-1. Hatched lines in Figs. a-d indicates a region of 95% confidence level based on the student's t-test. (Figure created using the COLA/GrADS software).

Figures 2a-b show the temporal evolution of vertical ozone anomalies averaged over
South Asia for the 2018 split-dSSW event and the composite of all split-dSSWs. There is a



large ozone enhancement in the UTLS, with values >80% (>150 ppb) in 2018 and >30%
(>80 ppb) for the composite of split-dSSWs within ±6 days around the SSW-onset. Figure
2a-b indicates that the ozone enhancements in the UTLS region are episodic and coincide
with downward propagating negative geopotential height (GPH) anomalies. The negative
GPH anomaly and lowering of the 380 K potential temperature isoline along with a positive
ozone enhancement in Fig. 2a-b indicate stratospheric intrusions associated with SSW events.
The ozone intrusions over the Indian region are stronger from 5 days before the onset and last
for ~15 days, causing an ozone enhancement of ~36% in 2018 and 16 % in split-DSSW
composite in the UTLS during this period (see Fig. S5a-b).

Figure 2 a-b shows that ozone enhancement in the UTLS is smaller in composite split-

dSSW than in 2018 (in the UTLS, and surface). This subdued effect is due to averaging
across multiple episodic events occurring at different times within ±30 days around the SSW
onset. Hence, to show the ozone enhancement during the SSW event, we picked up the
maximum ozone increase within ±30 days in the upper troposphere and near the surface over
South Asia for each of the split dSSWs (Fig. 2c-d). Figure 2c-d shows clear evidence of a
substantial ozone increase (50 to 250 %) in the UTLS (150 hPa) and (15 to 45%) near the
surface (850 hPa) during the split-dSSWs. Further, the lead-lag correlation between the ozone
variation in the upper troposphere and at surface levels shows that downward propagation of
ozone from 200 hPa to the near-surface occurs with 10- and 25-days lag in 2018 and 5- days
lead to 10- days lag in the case of composite (Fig. S6).







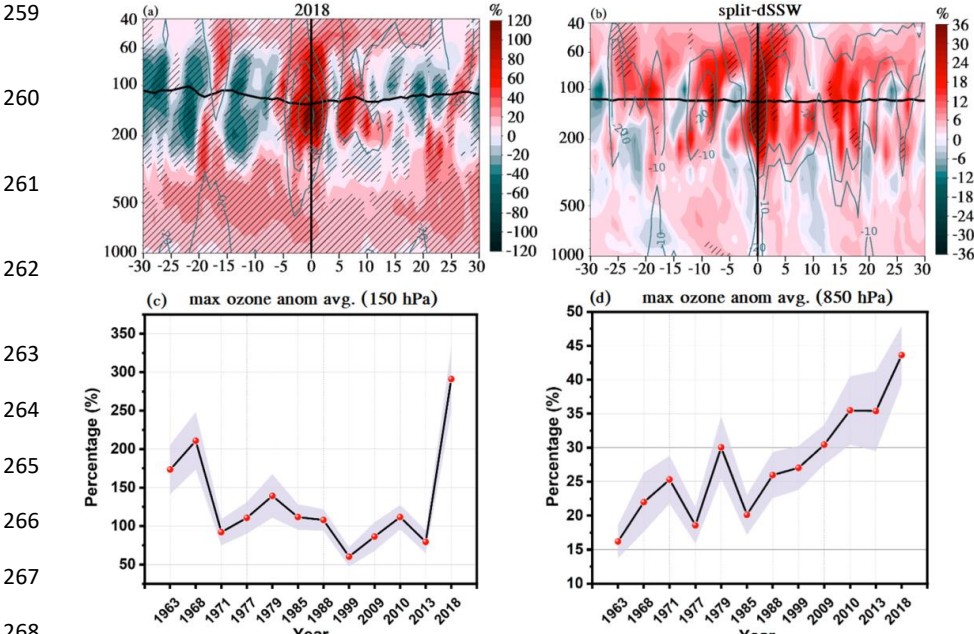

**Figure 2.** Temporal evolution of vertical ozone anomalies averaged over the South Asian region (65-90°E, 20-35°N) from 30 days before to 30 days after the onset for (a) the 2018 split dSSW event and (b) all the split dSSWs. Hatched lines in Figs. a-b indicates a region of 95% confidence level based on the student's t-test. The sky blue contour line represents the GPH anomaly during the respective period. The horizontal solid line represents 380 K potential temperature isoline, and the vertical solid line represents the onset day. Average of the daily maximum ozone increase within ±30 days over South Asia for each of the split dSSWs in the (c) upper troposphere (150 hPa) and (d) near-surface (850 hPa). The shading in (c) and (d) represents standard error. (Figure created using the COLA/GrADS software).

The latitude-pressure (Fig. 3a-b) and longitude-pressure (Fig. 3c-d) cross-sections of

ozone anomalies show large ozone enhancement for ±6 days around the onset in the UTLS

over South Asia exceeding >60% in 2018 and >20% in the composite of split-dSSW

events. Interestingly, a peak in ozone enhancement is seen at the subtropical jet core (Fig.

3a-b). This suggests the role of the subtropical jet causing ozone enhancement in the upper

troposphere over South Asia (discussed later in this section). The anomalous lowering of the

tropopause levels along with a strong negative GPH anomaly (indicating a low-pressure area)

coincident with large ozone enhancements, provides evidence of stratospheric intrusions

occurring during these split-dSSWs (Fig. 3c-d). Past literature reports ozone enhancements in





the polar region associated with dSSWs (e.g., Baldwin et al., 2021); however, high ozone

enhancement in the UTLS over the South Asian region underscores the unique regional

impacts of split-dSSWs.

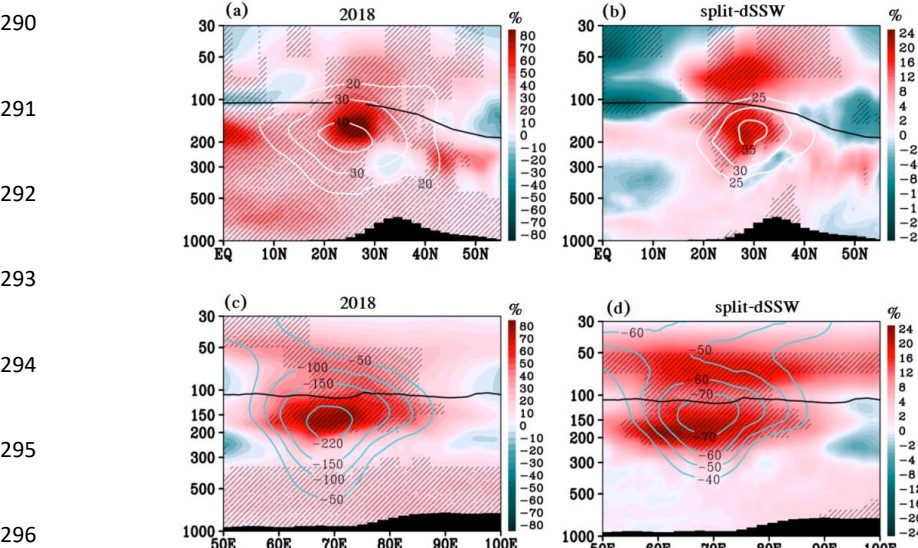

**Figure 3.** Latitude-pressure section of ozone anomalies averaged over South Asia (65 - 90°E)
for ±6 days around the split-dSSW onset for (a) the 2018 event, and (b) the composite of all
the split-dSSWs. Figures (c) and (d) are the same as those of (a) and (b) but represent
longitude variations of vertical ozone anomalies averaged over South Asia (20-35°N). White
contour lines in (a) and (b) represent the mean zonal wind, and sky blue contour lines in (c)
and (d) represent the GPH anomaly. Solid black lines in panels a-d represent the tropopause.
Hatched lines in panels a-d indicate a region of 95% confidence level based on the student's
t-test (Figure created using the COLA/GrADS software).

**3.2 Changes in upper tropospheric dynamics and Rossby wave activities associated with**

**the 2018 dSSW event**

In this section, we discuss the possible mechanism responsible for the ozone

enhancement in the UTLS over South Asia associated with the 2018 dSSW event. The
dynamic changes happening in the vortex for 2018 are discussed in the section 2. For the
2018 SSW, the vortex starts deforming 6 days before the SSW-onset and splits on 12
February 2018. After the vortex split, two baby vortices are seen centred over Eurasia and





North America for 6 days after the onset. Although the vortex over North America remains
anchored in that location, the vortex centred over Eurasia gradually drifts westward toward
the North Atlantic during this period (see section 2).

Several studies have shown that SSW-related planetary wave disturbances occur

across a deep layer of the stratosphere (e.g., McIntyre, 1982; McIntyre and Palmer, 1983;
Albers et al., 2016). These disturbances, which are strong in the mid-to-upper stratosphere,
extend downward and disrupt horizontal flows in the upper troposphere (200 hPa) (Albers et
al., 2016). We analyzed GPH anomalies at 10 hPa and 200 hPa to understand the coupling
between the stratosphere and the upper troposphere. Figure 4 illustrates the evolution of GPH
anomalies at 10 hPa and 200 hPa, and ozone anomalies at 200 hPa for ±6 days around the
SSW onset (Feb 12, 2018). Our analysis shows strong vertical coherence between 10 and 200
hPa levels (see Fig. 4a–e and Fig. 4f–j). Interestingly, the GPH anomalies at 10 hPa and 200
hPa north of 40°N show patterns of wave-1 and wave-2 with lows over America and Eurasia
for ±6 days around the onset. The SSW event thus affects the upper tropospheric subtropical
jet, which peaks at 200 hPa (Albers et al., 2016).

When the polar vortex splits and the baby vortices move towards the mid-latitudes,

they push the mid-latitude synoptic-scale waveguide structure farther equatorward. This
displacement channels the eastward propagating synoptic-scale Rossby waves toward the
Indian Ocean region, where they eventually break (Albers et al., 2016). The pattern of high
and low GPH anomalies in the subtropical region (15-40°N) seen in Fig. 4f-j shows synoptic-
scale Rossby waves occurring in the upper troposphere. Rossby wave breaking (RWB) is
seen as large filaments of high-PV air extending towards the equator. Such intrusions extend
downward from the lower stratosphere into the upper troposphere, causing an enhancement in
ozone (e.g. Holton et al., 1995; Waugh and Polvani, 2000; Albers et al., 2016). The 2 PVU



contour lines and ozone anomaly maps at 200 hPa depicted in Fig. 4k-o show clear
indications of ozone intrusions penetrating deep into the tropics, particularly over South Asia.
The persistent low and high GPH anomaly over South Asia indicates a deepening trough
associated with the eastward propagation of Rossby waves, facilitating enhanced
stratospheric intrusions (Figs. 4f-j).

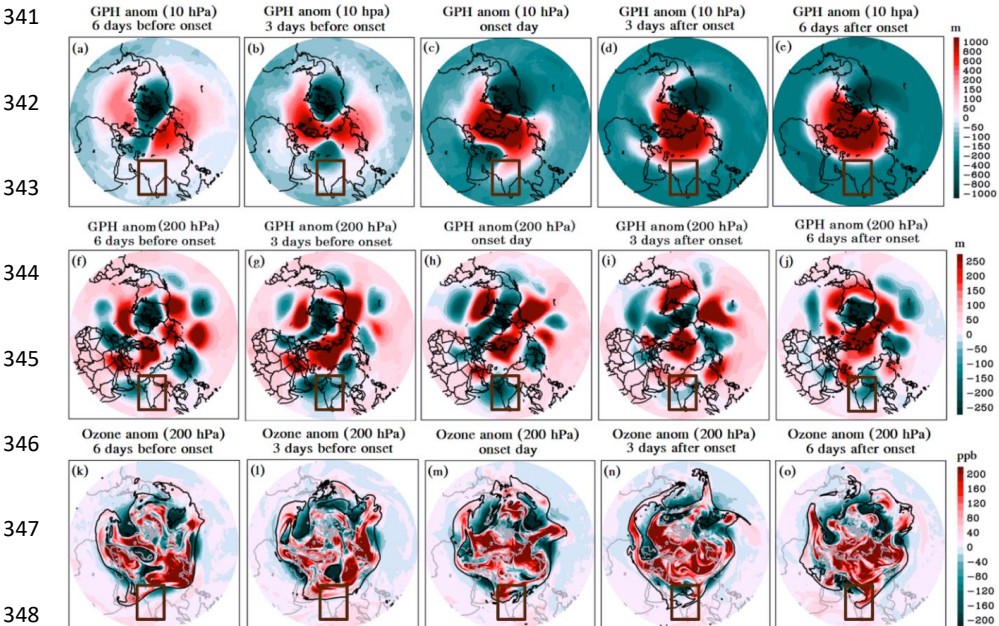

**Figure 4.** Spatial map of (a-e) GPH anomaly at 10 hPa, (f-j) GPH anomaly at 200 hPa, and
(k-o) ozone anomaly at 200 hPa from 6 days before to 6 days after the onset of the 2018 split-
dSSW event, shown at 3-day intervals. The black solid line in panels (k-o) represents the 2
PVU isoline. The square box represents the South Asian region considered for the present
study (Figure created using the COLA/GrADS software).

RWB facilitates the stripping of stratospheric air (indicated by the 2 PVU contour)
along the eastern flank of an anticyclonic centre (positive GPH anomaly, Fig. 4f–j) over the
South Asian region, causing large ozone enhancements (Fig. 4k–o). Figures 4k-o clearly
show that these episodic intrusions cause large ozone enhancements >150 ppb (>80%) over
South Asia. The time evolution of zonal winds depicted in Figure 5a-b also shows that thirty





days before the onset, the subtropical jet core is positioned over the northern part of the

Indian subcontinent but migrates equatorward near the onset day (Fig. 5a). The vertical

variation of zonal wind clearly depicts the intensification of westerly wind over the Indian

region around 200 hPa close to the onset day, facilitating Rossby wave intrusions (Fig. 5b).

The strength of zonal wind is strong within ±6 days around the onset, causing higher ozone

intrusions during this period (see Figs. 2a-b and Fig. 5b). Earlier, Albers et al. (2016) have

shown that the PV intrusion associated with split SSWs has a significant contribution over the

Indian Ocean. Further, we analysed the synoptic wave structure prevailing in the upper

troposphere for different SSW cases viz. split-dSSW, split-nSSW, displaced-dSSW, and

displaced-nSSW (Figures 5c-f). Figure 5c clearly shows that the persistence of synoptic wave

structures is prominent only in the split-dSSW cases. It is not evident in other SSW types

(Fig. 5d-f). This persistence of synoptic wave structures enhances RWBs, causing large

ozone intrusions during the split-dSSW events.

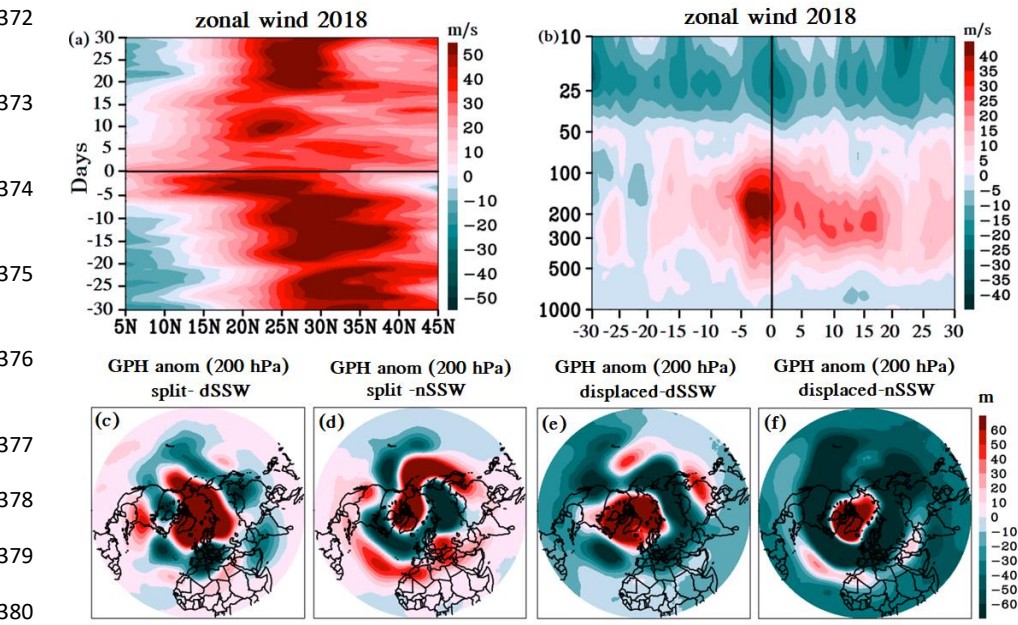




**Figure 5.** (a) Latitude-time plot of zonal wind averaged over South Asia (65° to 90° E). (b)
Temporal evolution of vertical zonal wind averaged over the South Asian region (65 - 90° E,
10 - 20° N) for ±30 days around the onset of the 2018 dSSW event. The horizontal (in Fig. a)
and vertical (in Fig. b) solid lines in the figure represent the onset day. Composite map of
GPH anomaly at 200 hPa averaged for ±6 days around the onset during (c) split-dSSW, (d)
split-nSSW, (e) displaced-dSSW, and (f) displaced-nSSW cases. (Figure created using the
COLA/GrADS software).
**3.3 Radiative impact of ozone associated with split-dSSW over the South Asian region**
Further, we assessed the radiative impact of ozone enhancements in the UTLS and
troposphere over the South Asian region associated with split-dSSW events. Figure 6
presents the computed radiative forcing of ozone averaged over the South Asian region for
the 2018 split-dSSW event and the composite of all downward propagating split-dSSWs for
±6 days around the onset. In the troposphere, a positive radiative forcing of $0.3 \pm 0.1$ W.m$^{-2}$ is
observed in the 2018 split-dSSW event, while the composite exhibits a forcing of $0.04 \pm 0.03$
W.m$^{-2}$ (Fig. 6). In contrast, the UTLS, where the largest percentage increases in ozone are
observed, shows a radiative forcing of $0.4 \pm 0.2$ W.m$^{-2}$ for the 2018 event and $0.08 \pm 0.06$
W.m$^{-2}$ for the composite. For the total atmosphere, the ozone radiative forcing is $0.5 \pm 0.2$
W.m$^{-2}$ for the 2018 event and $0.1 \pm 0.06$ W.m$^{-2}$ for the composite. These results highlight the
significant role of split-dSSW events in modulating the radiative balance in the troposphere,
particularly in the UTLS over South Asia.

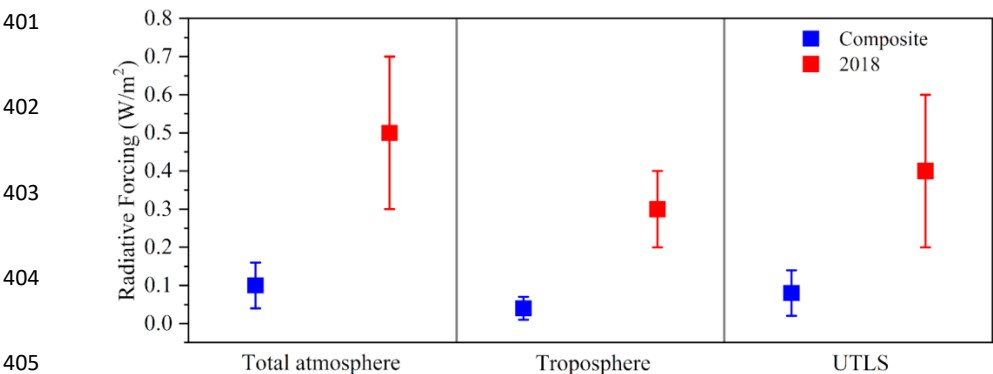



**Figure 6**. Average ozone radiative forcing (W.m$^{-2}$) in the total atmosphere, troposphere, and UTLS over the South Asian region calculated for the ±6 days around the onset of the 2018 SSW event and all split dSSW composites (Figure created using Origin, OriginLab, Northampton, MA).

**4. Conclusions**

The ERA5 reanalysis data shows Rossby wave intensification during split-dSSWs observed for the 1962-2018 time period, which lead to upper tropospheric intrusions over the South Asian region. This leads to substantial ozone enhancements over South Asia, with maximum increases of ~290% and an average of 130% in the UTLS region, and surface-level enhancements reaching up to 43%, with an average increase of 27% (see Fig. 2c-d). The ozone enhancement due to split-dSSWs events is episodic, however, impacts are observed for ~2 months around the SSW onset date.

Near surface ozone enhancements over South Asia also result from factors other than SSW events, including (1) anthropogenic emissions, which increase surface ozone by 20–30 ppb over this region (Gao et al., 2020), and (2) biomass burning that is prominent in the winter and pre-monsoon period causing a surface ozone increase by ~5–10 ppb(Gao et al., 2020). However, ozone enhancements during split-dSSW are substantial over South Asia, leading to an increase in tropospheric ozone radiative forcing of 0.04 ± 0.03 W.m$^{-2}$ and in the UTLS by 0.08±0.06 W.m$^{-2}$ for the composite of all the split-dSSWs, in comparison, changes in anthropogenic emissions over South Asia contribute to a tropospheric ozone radiative impact by 0.02-0.04 W.m$^{-2}$ for 2013–2017 relative to 1995–1999(Wang et al., 2022). These findings underscore the critical role of split-dSSWs in modulating radiative forcing, highlighting their importance as natural drivers of climate variability in addition to anthropogenic influences.



This large increase in radiative forcing produces positive feedback on the warming

climate of this region. The ozone intrusions are warranted to elevate pollution effects and
climate warming, impacting people's health, the ecosystem, and the economy. It is projected
that the frequency of SSWs will increase in a warming climate, which will further increase
stratospheric ozone intrusions and potentially amplify the consequences of positive feedback
mechanisms. Hence, we emphasise that climate model should be extended to the stratosphere
including polar vortex dynamics for accurate prediction of climate over South Asia. The
increase in ozone levels due to biomass burning and anthropogenic activities in South Asia
during the winter and pre-monsoon seasons, combined with ozone enhancement from SSW
events, will exacerbate ozone pollution across the region. Since SSWs cause large increase in
tropospheric ozone over South Asia it should be considered as one of the predictors while
prediction of pollution.

**Code and data availability**
The code and data used in this paper are available from https://zenodo.org/uploads/14604205
**Author contributions**
Conceptualisation: S.F. Supervision: SF, MH, PH, RF Investigation and methodology: SC,
SR and VS. Writing: all authors.
**Competing interests**
At least one of the (co-)authors is a member of the editorial board of Atmospheric Chemistry
and Physics.






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
