# Peer review of "Large Ozone Intrusions during Sudden Stratospheric Warmings Enhance Ozone Radiative Forcing over South Asia Shubhajyoti Roy1, Satheesh Chandran PR1, Suvarna Fadnavis1\*, Vijay Sagar1, Michaela I. Hegglin2, Rolf Müller2 1Centre for Climate Change Research, Indian Institute of Tropical Meteorology, India 2Institute of Energy and Climate Systems: Stratosphere (ICE-4), Forschungszentrum, Jülich, Germa"

_EGUsphere, 2025_

## Referee Comment (RC2)

**Review of egusphere-2025-1098:**

**Large Ozone Intrusions during Sudden Stratospheric Warmings Enhance Ozone Radiative Forcing over South Asia by S. Roy et al.**

The study of Roy et al. presents an analysis on how sudden stratospheric warmings (SSWs) related to a split vortex potentially affect ozone levels in the troposphere over South Asia. The use ERA5 data and categorize all SSWs between 1963 and 2018 into split vortex and displaced vortex events, as well as in downward and non-downward propagating SSWs. Furthermore, they discuss an event from 2018 in more detail and also present a composite analysis on all downward propagating, split vortex SSWs. From their point of view they conclude that ozone is enhanced due to Rossby wave dynamics in relation to the SSWs which in turn affects the ozone levels in the troposphere over South Asia and enhances the tropospheric radiative forcing from ozone.

The impact of SSWs on the troposphere has been documented in recent literature. The local effect of changes in trace gas concentrations, here particularly, on additional ozone from the stratosphere in the troposphere is still an open question. For this the topic is of relevance and also in scope of Atmospheric Chemistry and Physics. However, the conclusions in the present study look not very convincing to me so far. I have issues following the discussion in the present form. In particular, I feel that the connection between the SSW and the tropospheric effects needs more in depth analysis, or at least more convincing arguments in written form. Currently, I often do not see the connection between the SSW and tropospheric effects beyond vague lines of argumentation. I therefore recommend major revision before publication. I will lay out my concerns in more detail below and I hope that the authors can resolve my concerns in a revised manuscript.

**Major comments**

1) **Presentation of case study in 2018**
   In Section 2.3 the case study is introduced. This is an important point because here the idea of the study is presented and the basis is laid for the composite analysis. I would therefore recommend to discuss this case in more detail in an individual section. The figures should also be part of the main manuscript and not the supplement. More so, a reason should be given why this case is presented in more detail and not one of the other 11 dSSW cases.
   I would also recommend to put all figures related to this case into this discussion first. And then have a separate discussion of the composite, i.e., Figures 1a, 2a, 3a,c, 4, 5a,b.

2) **Composites**
   The composites are made from a very small number of events which is of course related to the fact that the discussed feature is a rare event. But of course this makes the composites also susceptible to outliers. In this case the 2018 case looks like an outlier (in particular, this seems to be the case in the Fig. 3c). So I wonder how much does the 2018 case contribute to the shape of the composite? Or vice versa, how does the composite look like without the 2018 case? In particular, Figures 3a and 3b look very much alike and made me wonder about this.

3) **Connection between stratospheric and upper tropospheric dynamics**
   I have issues with the chronological sequence of the processes. In Figure 2a,b the

ozone maxima in the UTLS occur right at the time of the SSW event. But should there not be a time lag between the vortex split at around 10 hPa and the effects evident at 200 hPa? In line 322 it is stated: *"Our analysis shows strong vertical coherence between 10 and 200 hPa levels (see Fig. 4a–e and Fig. 4f–j)"* But actually, I do not see a vertical coherence in these figures. I also would not expect it due to the time lag between the processes at 10 and 200 hPa.

What I also wonder is why the maximum ozone anomaly is evident even before the onset of the split dSSW. The pattern in Fig 2a looks for me more like a positive anomaly caused by RW dynamics. But from this figure I do not directly see the connection to the split dSSW. I think the authors should work out this point much clearer.

Also the discussion centered around Fig 4 f)-o) and Fig. 5 looks to my like a discussion which is centered around Rossby wave dynamics. Anomalies in GPH in Fig. 4 f)-j) show positive and negative anomalies related to a Rossby wave train and Fig. 4 k)-o) show the associated ozone. In Fig. 5a) a strong jet is evident and in Fig 5 b) it becomes evident that the jetstream maximum is located over South Asia during the time of the onset of the dSSW. But in all this discussion I do not see the connection to what is happening in the mid- to upper stratosphere. I would ask the authors to better show the connection between the Rossby wave dynamics in the UTLS with the vortex in the stratosphere in a clearer way.

4) **Impact on troposphere**
The first point I would like to make here is that is not once shown that the ozone anomalies at 200 hPa are undergoing stratosphere-troposphere exchange. From Fig. 4 k)-o) I would rather argue that the anomalies are all on the stratospheric side of the tropopause and thus have not really a significant impact on the troposphere. At least for the 2018 case study the authors should try to assess the related ozone flux from the stratosphere into the troposphere (e.g., using trajectories to calculate a mass flux, see Skerlak et al., 2014: https://doi.org/10.5194/acp-14-913-2014). The ozone impact which is discussed in the paper is more built on the ozone related transport within Rossby waves which simply advect stratospheric air masses over South Asia which then at 200 hPa produces a positive ozone anomaly. However, without an assessment of the ozone flux into the troposphere this does not significantly affect the tropospheric ozone concentration.

I also find Fig. 2d) not very convincing. The maximum ozone at 850 hPa shows to me rather the near surface increase in pollution levels over South Asia over the years. Again I am missing the connection to the stratospheric dynamics here.

I want to make clear here, I do not say that there is no ozone flux from the stratosphere into the troposphere but I do not see any proof yet that ozone transport takes place in relation to the dSSW in the presented analysis.

In turn, this puts the entire discussion centered around the radiative impact into question since I can not say whether the authors really determine the effect from ozone transported into the troposphere.

**Minor comments/technical comments: (in order of appearance)**

- Line 25: ERA-5 → ERA5
- Line 129: here you state +/- 61 days but later it is always stated +/- 60 days for the composites
- Line 141: Why are temperature, water vapor and clouds taken from the ECMWF forecast and not from ERA5?
- Line 146: The agreement between the radiative kernel technique and the radiative transfer model is given globally. But this study looks at local effects, so what are the maximum and minimum differences between these methods. What does a radiative transfer model take into account what the radiative kernel technique does not?
- Line 147/148: The ERA5 data is interpolated onto the kernel resolution for which it is specified to have 60 levels. How are this levels distributed in the atmosphere and what is the vertical resolution?
- Line 152: UTLS has not yet been defined (except in the abstract)
- Line 155 ff: Is it possible to include a satellite ozone product in the 2018 case study? This might help in the "validation" of the technique. Even more because the analysis is heavily based on ERA5 ozone.
- Line 207: couple → couples
- Line 232: Can you explain more on which data exactly you applied the student's t-test? Some assumption must be made to apply this test where I see potential issues (normality of data, continuous data, ….)
- Line 243: DSSW → dSSW
- Line 333ff: How does a filament extend downward from the LS into the UT? A filament is first of all a quasi-isentropic equatorward excursion of a stratospheric air mass. It is "downward" in a sense that the lower tropopause from high latitudes also moves toward the equator.
- Line 354ff and Figure 4: It is not easy to relate this discussion to the relevant parts in the figures. Maybe it is worth splitting the figure and increase the individual panel sizes to better highlight the features the authors need for the discussion.
- Figure 5a and related discussion: I do not know what the take away message here is. In Figure 5a, we simply see that there is a jet which is slightly weaker after the onset of the SSW. But is this already the effect of the SSW? At which altitude is this Hovmuller diagram taken?
- Figure 5b: Again how is this related to the vortex, all I see is that there is a maximum around the SSW onset.
- Line 362: What do you mean with Rossby wave intrusion?
- Sect. 3.3: How do you compute the uncertainties in the radiative forcing which are given in the text? The radiative forcing which is given here, is this relative to the non-SSW climatology?

---

## Author Comment (AC1)

**Response to the reviewer-II**

Large Ozone Intrusions during Sudden Stratospheric Warmings Enhance Ozone Radiative Forcing over South Asia by S. Roy et al. The study of Roy et al. presents an analysis on how sudden stratospheric warmings (SSWs) related to a split vortex potentially affect ozone levels in the troposphere over South Asia. The use ERA5 data and categorize all SSWs between 1963 and 2018 into split vortex and displaced vortex events, as well as in downward and non-downward propagating SSWs. Furthermore, they discuss an event from 2018 in more detail and also present a composite analysis on all downward propagating, split vortex SSWs. From their point of view, they conclude that ozone is enhanced due to Rossby wave dynamics in relation to the SSWs which in turn affects the ozone levels in the troposphere over South Asia and enhances the tropospheric radiative forcing from ozone.

The impact of SSWs on the troposphere has been documented in recent literature. The local effect of changes in trace gas concentrations, here particularly, on additional ozone from the stratosphere in the troposphere is still an open question. For this the topic is of relevance and also in scope of Atmospheric Chemistry and Physics. However, the conclusions in the present study look not very convincing to me so far. I have issues following the discussion in the present form. In particular, I feel that the connection between the SSW and the tropospheric effects needs more in depth analysis, or at least more convincing arguments in written form. Currently, I often do not see the connection between the SSW and tropospheric effects beyond vague lines of argumentation. I therefore recommend major revision before publication. I will lay out my concerns in more detail below and I hope that the authors can resolve my concerns in a revised manuscript.

**Response:** We sincerely thank the reviewer for the meticulous evaluation, constructive comments, and valuable suggestions. We acknowledge the concern regarding the clarity of the connection between SSWs and tropospheric impacts in the current version. Our revised analysis reveals that, phase of the Quasi-Biennial Oscillation (QBO) plays a more critical role in shifting the subtropical jet position and associated Rossby wave breaking over the South Asian region during SSWs. Therefore, in the revised manuscript, we have reclassified the SSWs into two categories based on the prevailing QBO phase: SSWs coinciding with the westerly phase (WQBO-SSW) and those coinciding with the easterly phase (EQBO-SSW). Our results show that WQBO-SSW events are associated with more ozone intrusions compared to EQBO-SSW events. Accordingly, we have revised sections 3.1 and 3.2.

**Major comments**
1) Presentation of case study in 2018 In Section 2.3 the case study is introduced. This is an important point because here the idea of the study is presented and the basis is laid for the composite analysis. I would therefore recommend to discuss this case in more detail in an individual section. The figures should also be part of the main manuscript and not the

supplement. More so, a reason should be given why this case is presented in more detail and not one of the other 11 dSSW cases. I would also recommend to put all figures related to this case into this discussion first. And then have a separate discussion of the composite, i.e., Figures 1a, 2a, 3a,c, 4, 5a,b.

**Response (1):** As suggested by the reviewer, a separate section is added for the 2018 SSW event (Sec. 3.1), followed by the discussion of composite (Sec. 3.2) in the revised manuscript. Also, we have now included the motivation for taking the 2018 SSW as a case study in the introduction (L97-100).

The supplementary figures (wind reversal (Fig. S1) and PV evolution (Fig. S2)) are provided to show the change in stratospheric dynamics around the onset of the 2018 SSW event. However, the revised analysis now emphasizes the role of the QBO in modulating the equatorward shift of the subtropical jet and associated ozone intrusion. Therefore, including those supplementary figures in the main text will not add substantial new information to the ongoing discussion. Hence, we have not moved those figures from the supplement.

2) Composites
The composites are made from a very small number of events which is of course related to the fact that the discussed feature is a rare event. But of course this makes the composites also susceptible to outliers. In this case the 2018 case looks like an outlier (in particular, this seems to be the case in the Fig. 3c). So I wonder how much does the 2018 case contribute to the shape of the composite? Or vice versa, how does the composite look like without the 2018 case? In particular, Figures 3a and 3b look very much alike and made me wonder about this.

**Response (2):** We understand the reviewer's concern. We have now shown the composite excluding the 2018 event (Fig. 4 in the revised manuscript). The figure below shows the composite excluding the 2018 event for your reference. Although the values have marginal differences, the pattern remains largely the same.

[Figure]

Figure 1: (a) Temporal evolution of vertical ozone anomalies averaged over South Asia (65-90°E, 20-35°N) from ±30 days around the onset for the composite of all WQBO-SSWs without 2018. (b) Latitude-pressure section of ozone anomalies averaged over South Asia (65-90°E) for ±6 days around the onset for the composite of WQBO-SSWs without 2018.

3) Connection between stratospheric and upper tropospheric dynamics
I have issues with the chronological sequence of the processes. In Figure 2a,b the ozone maxima in the UTLS occur right at the time of the SSW event. But should there not be a time lag between the vortex split at around 10 hPa and the effects evident at 200 hPa? In line 322 it is stated: "Our analysis shows strong vertical coherence between 10 and 200 hPa levels (see Fig. 4a–e and Fig. 4f–j)" But actually, I do not see a vertical coherence in these figures. I also would not expect it due to the time lag between the processes at 10 and 200 hPa. What I also wonder is why the maximum ozone anomaly is evident even before the onset of the split dSSW. The pattern in Fig 2a looks for me more like a positive anomaly caused by RW dynamics. But from this figure I do not directly see the connection to the split dSSW. I think the authors should work out this point much clearer.
Also the discussion centered around Fig 4 f)-o) and Fig. 5 looks to my like a discussion which is centered around Rossby wave dynamics. Anomalies in GPH in Fig. 4 f)-j) show positive and negative anomalies related to a Rossby wave train and Fig. 4 k)-o) show the associated ozone. In Fig. 5a) a strong jet is evident and in Fig 5 b) it becomes evident that the jetstream maximum is located over South Asia during the time of the onset of the dSSW. But in all this discussion I do not see the connection to what is happening in the mid- to upper stratosphere. I would ask the authors to better show the connection between the Rossby wave dynamics in the UTLS with the vortex in the stratosphere in a clearer way.

**Response (3):** We acknowledge the reviewer's concern. However, we would like to emphasise that, unlike high latitudes, where SSWs exert a direct downward influence on the troposphere, our analysis indicates that the low-latitude responses (such as over South Asia) are mediated primarily by Rossby-wave dynamics. Specifically, by RWB and PV streamer intrusions along the subtropical waveguide. The location of these RWBs are modulated by the positioning of the subtropical jet during SSWs. It is well established that major SSWs are preceded by enhanced mid-latitude planetary and synoptic wave driving (e.g., Baldwin et al., 2021). Whether and how that wave activity projects into South Asia depends on the background flow set by the QBO. During the westerly QBO, the associated secondary circulation warms the equatorial lower stratosphere and cools the subtropics, sharpening and shifting the UTLS meridional temperature gradient equatorward (e.g., Hitchman et al., 2021). By thermal-wind balance, this strengthens upper-tropospheric westerlies on the equatorward flank and displaces the subtropical jet equatorward over the South Asain longitudes, favouring subtropical wave guidance, RWB, and PV-streamer intrusions. Accordingly, because the ozone enhancement is mediated by these RWB/PV-streamer intrusions, features that are often established prior to SSW onset, we do not expect a time lag. We now detail this mechanism in Section 3.1.1.

To the reviewer's comment on "vertical coherence", the reviewer may kindly note that in Figure 4, we observed that regions with low GPH at 10 hPa (Fig. 4a-e in the old manuscript) correspond to similar low GPH anomalies at 200 hPa (Fig. 4f-j in the old manuscript). To describe this feature, we used the term 'strong vertical coherence'. However, in the revised manuscript, we have removed the term 'strong vertical coherence' and Figure 4(a-e) (in the old manuscript).

(a) Baldwin, M. P., Domeisen, D. I. V., Hegglin, M. I., Garny, H., Garfinkel, C. I., Langematz, U., Charlton-Perez, A. J., Butchart, N., Gerber, E. P., Birner, T., Butler, A. H., Ayarzagüena, B., and Pedatella, N. M.: Sudden Stratospheric Warmings, Reviews of Geophysics, 59, https://doi.org/10.1029/2020rg000708, 2021.
(b) Hitchman, M. H., Tegtmeier, S., Yoden, S., Haynes, P. H., and Kumar, V.: An Observational History of the Direct Influence of the Stratospheric Quasi-biennial Oscillation on the Tropical and Subtropical Upper Troposphere and Lower Stratosphere, Journal of the Meteorological Society of Japan. Ser. II, 99, 239–267, https://doi.org/10.2151/jmsj.2021-012, 2021.

4) Impact on troposphere

The first point I would like to make here is that is not once shown that the ozone anomalies at 200 hPa are undergoing stratosphere-troposphere exchange. From Fig. 4 k)-o) I would rather argue that the anomalies are all on the stratospheric side of the tropopause and thus have not really a significant impact on the troposphere. At least for the 2018 case study the authors should try to assess the related ozone flux from the stratosphere into the troposphere (e.g., using trajectories to calculate a mass flux, see Skerlak et al., 2014: https://doi.org/10.5194/acp-14-913-2014). The ozone impact which is discussed in the paper is more built on the ozone related transport within Rossby waves which simply advect stratospheric air masses over South Asia which then at 200 hPa produces a positive ozone anomaly. However, without an assessment of the ozone flux into the troposphere this does not significantly affect the tropospheric ozone concentration.

I also find Fig. 2d) not very convincing. The maximum ozone at 850 hPa shows to me rather the near surface increase in pollution levels over South Asia over the years. Again I am missing the connection to the stratospheric dynamics here.

I want to make clear here, I do not say that there is no ozone flux from the stratosphere into the troposphere but I do not see any proof yet that ozone transport takes place in relation to the dSSW in the presented analysis.

In turn, this puts the entire discussion centered around the radiative impact into question since I can not say whether the authors really determine the effect from ozone transported into the troposphere.

**Response (4):** We acknowledge the reviewer's concern. As shown in Figure 3a of the previous manuscript (now Fig. 1c), the maximum ozone enhancement during the 2018 event occurs below the tropopause, particularly in the upper troposphere near 200 hPa. To highlight this, Figure 4(k–o) (now Fig. 2f–j) focuses on ozone anomalies at 200 hPa, illustrating that the enhancements are primarily in the upper troposphere rather than confined to the stratosphere.

To the reviewer's comment on "ozone flux from the stratosphere.", the reviewer may kindly note that a full flux attribution using trajectory analysis is beyond the scope of the present study. However, to attribute the upper tropospheric ozone enhancement over South Asia to a stratospheric source, we use potential vorticity (PV) as a dynamical tracer (Fig. 2f-j and Fig. 6). PV values ≥ 2 PVU are widely used to delineate intrusions of stratospheric air into the upper troposphere (Holton et al., 1995; Kunz et al., 2015). Furthermore, we observed that negative geopotential height (GPH) anomalies in the UTLS correspond to positive ozone anomalies over the study region (Fig. 1a and 4a). This indicates troughing over the region and is commonly associated with enhanced stratospheric influence and higher ozone in the UTLS (e.g., Steinbrecht et al., 1998; Albers et al., 2022; Chen et al., 2019). We have clarified these in the revised manuscript (Sec 3.1.1).

To the reviewer's comment on "maximum ozone at 850 hPa..", in the revised manuscript, after detrending the record (1962–2018), near-surface signals were weak and not statistically significant; accordingly, Fig. 2d and all surface ozone discussion have been removed from the revised manuscript.

**Minor comments/technical comments: (in order of appearance)**

5. Line 25: ERA–5 → ERA5

**Response (5):** Corrected in the revised manuscript (L21).

6. Line 129: here you state +/- 61 days but later it is always stated +/- 60 days for the Composites.

**Response (6):** Thank you for bringing this to our attention. We have harmonized the values to +/-30 days in the revised manuscript (L126-127).

7. Line 141: Why are temperature, water vapor and clouds taken from the ECMWF forecast and not from ERA5?

**Response (7):** The reviewer may kindly note that, in our study, we did not directly use ECMWF forecast data. Instead, we applied the radiative kernel developed by Skeie et al. (2020), which was constructed using monthly mean meteorological fields (temperature, water vapour, and clouds) from ECMWF. This kernel provides a consistent framework for estimating ozone

radiative forcing. In our analysis, we only used this kernel to compute the radiative forcing of ozone based on ERA5 ozone anomalies.

8. Line 146: The agreement between the radiative kernel technique and the radiative transfer model is given globally. But this study looks at local effects, so what are the maximum and minimum differences between these methods. What does a radiative transfer model take into account what the radiative kernel technique does not?

**Response (8):** We acknowledge the reviewer's concern. The radiative kernel (Skeie et al., 2020) has sufficient resolution (~5.6° × 5.6° horizontal, with 60 vertical levels) to calculate the RF over the study region (65-90°E, 20-35°N).

The radiative kernel method is computationally efficient for quantifying the radiative forcing than the radiative transfer model. The kernel is the partial derivative of the radiative flux calculated by a radiative transfer model with respect to small perturbations in ozone concentration at different altitudes. It is multiplied by the change in ozone concentration to compute the radiative forcing. While the radiative transfer model calculates the direct radiative transfer through the atmosphere, it considers the absorption, emission, and scattering by ozone.

9. Line 147/148: The ERA5 data is interpolated onto the kernel resolution for which it is specified to have 60 levels. How are this levels distributed in the atmosphere and what is the vertical resolution?

**Response (9):** The ERA5 data were linearly interpolated in pressure to 60 vertical levels, ranging from the surface up to 0.1 hPa, to match the vertical resolution of the radiative kernel. The vertical resolution of the kernel is unevenly distributed vertically. In the UTLS, it has 14 layers with level spacing ranging from 12 hPa to 34.5 hPa.

10. Line 152: UTLS has not yet been defined (except in the abstract)

**Response (10):** Thanks for pointing this out. We have defined UTLS in the revised manuscript

(L93).

11. Line 155 ff: Is it possible to include a satellite ozone product in the 2018 case study? This might help in the "validation" of the technique. Even more because the analysis is heavily based on ERA5 ozone.

**Response (11):** Thank you for the suggestion. Unfortunately, daily UTLS ozone profiles over our study region are limited: limb profilers (e.g., Aura MLS) sample too sparsely for sub-regional, day-to-day maps, and nadir sensors (e.g., OMI/OMPS) provide column ozone rather than UTLS-specific fields. We therefore rely on ERA5 ozone, which is produced by 4D-Var and assimilates multiple satellite and ground-based datasets (TOMS, SBUV/2 v8.6, SCIAMACHY,

MIPAS, Aura MLS, OMI), supporting its use here (Hersbach et al., 2020; S-RIP Final Report, 2022).

a) Hersbach, H., Bell, B., Berrisford, P., Hirahara, S., Horányi, A., Muñoz-Sabater, J., Nicolas, J., Peubey, C., Radu, R., Schepers, D., Simmons, A., Soci, C., Abdalla, S., Abellan, X., Balsamo, G., Bechtold, P., Biavati, G., Bidlot, J., Bonavita, M., De Chiara, G., Dahlgren, P., Dee, D., Diamantakis, M., Dragani, R., Flemming, J., Forbes, R., Fuentes, M., Geer, A., Haimberger, L., Healy, S., Hogan, R. J., Hólm, E., Janisková, M., Keeley, S., Laloyaux, P., Lopez, P., Lupu, C., Radnoti, G., de Rosnay, P., Rozum, I., Vamborg, F., Villaume, S. and Thépaut, J.: The ERA5 global reanalysis, Quarterly Journal of the Royal Meteorological Society, 146(730), 1999–2049, doi:10.1002/qj.3803, 2020.

b) SPARC Reanalysis Intercomparison Project (S-RIP) Final Report. M. Fujiwara, G.L. Manney, L.J. Gray, and J.S. Wright (Eds.), SPARC Report No. 10, WCRP-17/2020, doi: 10.17874/800dee57d13, available at www.sparc-climate.org/publications/sparc-reports, 2022.

•12. Line 207: couple → couples

**Response (12):** The sentence has been restructured in the revised manuscript.

13. Line 232: Can you explain more on which data exactly you applied the student's ttest? Some assumption must be made to apply this test where I see potential issues (normality of data, continuous data, ….)

**Response (13):** We apologize for not describing our statistical testing in sufficient detail. We have now added the details on how the statistical tests are done in the Methods section of the revised manuscript (L134-141).

Given the small sample size for and the likelihood of non-normal distributions, we replaced the initial Student's t-test with the Monte Carlo bootstrap and the Wilcoxon signed-rank test in the revised manuscript. "For the Monte Carlo, we built a calendar-matched null by resampling days from non-SSW years within the same day-of-year window. We then use a bias-corrected and accelerated (BCa) bootstrap with 20,000 resamples to form 95% confidence intervals. For 2018, we checked whether the observed value lay outside the BCa interval of the background ensemble. For the composite, we tested whether the mean anomaly differed from zero. Next, we applied an exact Wilcoxon signed-rank test to the same data. A grid point is called significant only when both tests agree at 95% significance." This approach offers three advantages: (a) distribution-free inference suitable for small samples, (b) improved coverage from BCa intervals that correct bias and skewness, and (c) robustness of the Wilcoxon test to outliers and non-

Gaussianity (Efron, 1987; Efron & Tibshirani, 1994; Davison & Hinkley, 1997; Wilcoxon, 1945).

(a) Efron, B.: Better Bootstrap Confidence Intervals, Journal of the American Statistical Association, 82, 171–185, https://doi.org/10.1080/01621459.1987.10478410, 1987.
(b) Efron, B. and Tibshirani, R. J.: An Introduction to the Bootstrap, chapman hall crc, https://doi.org/10.1201/9780429246593, 1994..
(c) Davison, A. C. and Hinkley, D. V.: Bootstrap Methods and their Application, cambridge university, https://doi.org/10.1017/cbo9780511802843, 1997.
(d) Wilcoxon, F.: Individual Comparisons by Ranking Methods, Biometrics Bulletin, 1, 80, https://doi.org/10.2307/3001968, 1945.

14. Line 243: DSSW → dSSW

**Response (14):** This term is replaced in the revised manuscript.

15. Line 333ff: How does a filament extend downward from the LS into the UT? A filament is first of all a quasi-isentropic equatorward excursion of a stratospheric air mass. It is "downward" in a sense that the lower tropopause from high latitudes also moves toward the equator.

**Response (15):** We thank the reviewer for this clarification. We agree with the interpretation and have revised the manuscript accordingly (L226-227)

16. Line 354 and Figure 4: It is not easy to relate this discussion to the relevant parts in the figures. Maybe it is worth splitting the figure and increase the individual panel sizes to better highlight the features the authors need for the discussion.

**Response (16):** Thanks for this suggestion. Figure 4 is modified in the revised manuscript (Fig. 2 in the revised manuscript). Figure 2 shows the 200 hPa GPH and PV contour to characterise the upper-tropospheric response to the vortex dynamics, together with ozone anomalies for ±6 days around SSW onset. This discussion has been added to the revised manuscript (Sec 3.1.1).

17. Figure 5a and related discussion: I do not know what the take away message here is. In Figure 5a, we simply see that there is a jet which is slightly weaker after the onset of the SSW. But is this already the effect of the SSW? At which altitude is this Hovmuller diagram taken?

**Response (17):** We acknowledge the reviewer's concern. Figure 5a (Fig. 3a in revised manuscript) is plotted at 200 hPa, where the subtropical jet core is most prominent over South Asia. Our revised analysis shows that the equatorward shift of the subtropical jet is more prominent during the westerly phase of QBO than the easterly phase (Fig. 5c in the revised manuscript) during SSW years. This equatorward shift creates favourable conditions for enhanced Rossby wave breaking and PV-streamer activity (Fig. 6 in the revised manuscript) over

the study region (e.g., Albers et al., 2016), thereby linking the jet dynamics to the subsequent stratospheric intrusions highlighted in our study. We have added this point in the revised manuscript (sec 3.1.1).

18. Figure 5b: Again how is this related to the vortex, all I see is that there is a maximum around the SSW onset.

**Response (18):** Figure 5b (in the old manuscript) illustrates the local wind response over the Indian sector around SSW onset, specifically, the transient intensification of the subtropical jet across UTLS levels. Our analysis suggests that the ozone enhancement observed over the South Asian region is more closely associated with Rossby-wave dynamics modulated by the SSW and concurrent QBO phase, rather than a direct downward influence of the polar vortex itself. Please see *Response (3)* for the detailed mechanism and citations.

19. Line 362: What do you mean with Rossby wave intrusion?

**Response (19):** We thank the reviewer for bringing this to our attention. In the revised manuscript, we have modified the wording to Rossby-wave breaking (L259).

20. Sect. 3.3: How do you compute the uncertainties in the radiative forcing which are given in the text? The radiative forcing which is given here, is this relative to the non-SSW climatology?

**Response (20):** We acknowledge the reviewer's concern. The radiative forcing values (mean ± 1 standard deviation) reported in our study are calculated relative to a baseline of non-SSW climatology conditions. The radiative forcing value was calculated as the spatial mean within the region for each event. The uncertainties are the ± 1 standard deviation within the region for each event.

---

## Author Comment (AC2)

**Replies to the reviewer-I**

Roy et al. analyze in their work ERA5 ozone data for different sudden stratospheric warmings in the northern hemisphere and connect these events with increased ozone concentrations at UTLS levels and near-surface. Finally, a brief analysis for the radiative impact is shown.

The manuscript addresses an important and interesting topic well within the scope of ACP, but I think this manuscript needs major revisions before considering publication. In particular, the main focus of this paper is not clear to me and also, it could be better structured. Another very important point to me is that the ERA5 data, which is the heart of this manuscript, is not discussed if it is suitable for this kind of analysis. Finally, the year 2018 is in focus without motivation why, but then its outstanding role in the time series plots is not discussed in detail. I see room for improvement on this manuscript.

**Response:** We sincerely thank the reviewer for the meticulous evaluation, constructive comments, and valuable suggestions. In the revised manuscript, we have incorporated all recommendations, clarified the main focus of the study, and improved the overall structure. We have also included a detailed discussion on the reliability of ERA5 ozone data for our analysis. Furthermore, we addressed the motivation for highlighting the 2018 SSW event and clarified the underlying mechanism. We greatly appreciate the reviewer's time and effort in helping to strengthen our work. All changes are marked in the track-change version of the manuscript, with line numbers referenced in the responses.

General comments:

1. Some figures are rather small and of poor quality. Please increase the size, in particular the font size of axes, labels, etc. Some axes or color bars are even not labeled at all: please add labels for all axes and color bars!

**Response (1):** Thank you for the helpful suggestion. As per the reviewer's suggestions, all figures have been modified (font sizes for axes, labels, and legends have been increased, and all missing axis and colour bar labels have been added) for better clarity and readability in the revised manuscript.

2. Many figures were produced using the COLA/GrADS software, but I did not find any information or reference to this tool. However, it is good that the authors mention how their figures are created.

**Response (2):** Thank you for pointing this out. In the revised manuscript, we regenerated all figures using Python and have explicitly stated this in the figure captions.

3. The analyses of this work are based on ERA5 ozone data. In order to estimate the findings of

this paper, I need some background information about the reliability of this ERA5 ozone data set. What is the base for the ERA5 ozone? Are there measurements assimilated? Are there validation papers? In particular: How is the UTLS represented in ERA5 in comparison to independent measurements? How does that change over the long time range analyzed in this work?

**Response (3):** We understand the reviewer's concern and thank you for pointing this out. ERA5 ozone is generated by 4D-Var assimilation and ingests multiple satellite and ground-based observations (e.g., TOMS, SBUV/2 v8.6, SCIAMACHY, MIPAS, Aura MLS, OMI) (Hersbach et al., 2020; S-RIP Final Report, 2022). In the revised manuscript, we have added validation/intercomparison studies of ERA5 ozone, particularly in the UTLS, to document its reliability. (L116-123).

To mitigate potential changes over the long-term range, we detrended the ozone time-series for 1962–2018 (as also suggested by Reviewer 3). This is now described in the Methods section of the revised manuscript (L132-134).

a) Hersbach, H., Bell, B., Berrisford, P., Hirahara, S., Horányi, A., Muñoz-Sabater, J., Nicolas, J., Peubey, C., Radu, R., Schepers, D., Simmons, A., Soci, C., Abdalla, S., Abellan, X., Balsamo, G., Bechtold, P., Biavati, G., Bidlot, J., Bonavita, M., De Chiara, G., Dahlgren, P., Dee, D., Diamantakis, M., Dragani, R., Flemming, J., Forbes, R., Fuentes, M., Geer, A., Haimberger, L., Healy, S., Hogan, R. J., Hólm, E., Janisková, M., Keeley, S., Laloyaux, P., Lopez, P., Lupu, C., Radnoti, G., de Rosnay, P., Rozum, I., Vamborg, F., Villaume, S. and Thépaut, J.: The ERA5 global reanalysis, Quarterly Journal of the Royal Meteorological Society, 146(730), 1999–2049, doi:10.1002/qj.3803, 2020.

b) SPARC Reanalysis Intercomparison Project (S-RIP) Final Report. M. Fujiwara, G.L. Manney, L.J. Gray, and J.S. Wright (Eds.), SPARC Report No. 10, WCRP-17/2020, doi: 10.17874/800dee57d13, available at www.sparc-climate.org/publications/sparc-reports, 2022.

4. Does ERA5 only show stratospheric ozone, which is transported to the UTLS, or are contributions from ground sources (anthropogenic, biomass burning,...) also included? If the tropospheric sources are included, how can these be separated from the stratospheric sources?

**Response (4):** The reviewer may kindly note that the ERA5 ozone is generated by assimilating multiple satellite and ground-based observations and gives ozone vertical profile from the surface up to a height of 80km (S-RIP Final Report, Hersbach et al. 2020).

To attribute the UTLS ozone enhancements in our study to a stratospheric source, we used potential vorticity (PV) as the primary dynamical tracer (Fig. 2f-j). PV values near or above ~2 PVU are widely used to delineate stratospheric air in the troposphere (Holton et al., 1995; Kunz

et al., 2015) (Sec. 2.1). Further, we noted that negative geopotential height (GPH) anomalies in the UTLS coincide with positive ozone anomalies over the study region (Fig. 1a and 4a). This indicates troughing over the region and is commonly associated with enhanced stratospheric influence and higher ozone in the UTLS (e.g., Steinbrecht et al., 1998; Albers et al., 2022; Chen et al., 2019). We have clarified these in the revised manuscript (Sec 3.1.1).

(a) Holton, J. R., Pfister, L., Haynes, P. H., Douglass, A. R., Rood, R. B., and Mcintyre, M. E.: Stratosphere-troposphere exchange, Reviews of Geophysics, 33, 403–439, https://doi.org/10.1029/95rg02097, 1995.

(b) Kunz, A., Wernli, H., and Sprenger, M.: Climatology of potential vorticity streamers and associated isentropic transport pathways across PV gradient barriers, Journal of Geophysical Research: Atmospheres, 120, 3802–3821, https://doi.org/10.1002/2014jd022615, 2015.

(c) Steinbrecht, W., Hoinka, K. P., Köhler, U., and Claude, H.: Correlations between tropopause height and total ozone: Implications for long-term changes, Journal of Geophysical Research: Atmospheres, 103, 19183–19192, https://doi.org/10.1029/98jd01929, 1998.

(d) Albers, J. R., Elsbury, D., Butler, A. H., Langford, A. O., and Breeden, M. L.: Dynamics of ENSO-driven stratosphere-to-troposphere transport of ozone over North America, Atmospheric Chemistry and Physics, 22, 13035–13048, https://doi.org/10.5194/acp-22-13035-2022, 2022.

(e) Chen, D., Ma, L.-Y., Zhou, T.-J., Guo, D., Shi, C.-H., and Chen, L.: Statistical Analysis of the Spatiotemporal Distribution of Ozone Induced by Cut-Off Lows in the Upper Troposphere and Lower Stratosphere over Northeast Asia, Atmosphere, 10, 696, https://doi.org/10.3390/atmos10110696, 2019.

5. In the description of the materials and methods, already results (e.g. from Fig. 1) are discussed, but the Figure is reintroduced again in Section 3.1. This makes the structure inconsistent and should be reconsidered.

**Response (5):** Thanks for pointing this out. Fig. S1 in the supplementary and its discussion in the methods section have now been removed in the revised manuscript to avoid redundancy and ensure a clearer structure.

6. The event in 2018 is discussed in detail in this manuscript. For that reason, I am missing some background information about this SSW. How was the Arctic winter 2017/18 in general prior to the SSW? Are there articles published dealing with this specific winter, or even this specific SSW? Further, Fig. 2c looks like that this 2018 event was an extreme outlier. Are there explanations for that outstanding behavior in the UTLS? Also, some motivation, why 2018 was chosen would be helpful.

**Response (6):** We have examined all major SSWs during the study period from 1962 to 2018. Our analysis reveals a relatively more equatorward shift of the subtropical jet over South Asia and a corresponding large ozone intrusion in this region during the 2018 SSW, compared to other SSW years. This motivated us to report the detailed mechanism of the 2018 SSW as a case study. We have included this motivation in the introduction section of the revised manuscript (L97-

100). This equatorward shift facilitates the eastward-propagating synoptic-scale Rossby waves to move further equatorward, favouring RWB and PV-stremer activity along with more ozone intrusion (Homeyer & Bowman, 2013; Albers et al., 2016). This unique dynamical setup explains why the 2018 event stands out as an outlier in the UTLS ozone response. In the revised manuscript, we have discussed this aspect in detail (see section 3.1.1).

To the reviewer's question "How was the Arctic winter 2017/18 in general prior to the SSW? and Are there articles published dealing with this specific winter, or even this specific SSW?", The Arctic winter preceding the February 2018 major SSW was anomalously warm and characterised by reduced sea-ice extent, with widespread positive surface-air temperature anomalies of ~4–7 °C (National Snow and Ice Data Centre report). The 2018 SSW and its mid-latitude impacts have been well-documented (e.g., King et al., 2019; Butler et al., 2020; Karpechko et al., 2018). A recent study by Shi et al. (2023) reported local temperature drops of 14-18°C over Eurasia and East Asia associated with the 2018 SSW. However, to our knowledge, UTLS ozone responses over South Asia during this event have received limited attention, which motivates our emphasis on the 2018 case.

(a) King, A. D., Rudeva, I., Jucker, M., Butler, A. H., and Earl, N. O.: Observed Relationships Between Sudden Stratospheric Warmings and European Climate Extremes, Journal of Geophysical Research: Atmospheres, 124, 13943–13961, https://doi.org/10.1029/2019jd030480, 2019.
(b) Butler, A. H., Lillo, S. P., Long, C. S., Lee, S. H., and Lawrence, Z. D.: Differences between the 2018 and 2019 stratospheric polar vortex split events, Quarterly Journal of the Royal Meteorological Society, 146, 3503–3521, https://doi.org/10.1002/qj.3858, 2020.
(c) Karpechko, A. Y., Tyrrell, N., Balmaseda, M., Charlton-Perez, A., and Vitart, F.: Predicting Sudden Stratospheric Warming 2018 and Its Climate Impacts With a Multimodel Ensemble, Geophysical Research Letters, 45, https://doi.org/10.1029/2018gl081091, 2018.
(d) Shi, Y., Evtushevsky, O., Milinevsky, G., Wang, X., Klekociuk, A., Han, W., Grytsai, A., Wang, Y., Wang, L., Novosyadlyj, B., and Andrienko, Y.: Impact of the 2018 major sudden stratospheric warming on weather over the midlatitude regions of Eastern Europe and East Asia, Atmospheric Research, 297, 107112, https://doi.org/10.1016/j.atmosres.2023.107112, 2023.

7. What is the focus of this paper with respect to ozone? Showing the contribution of stratospheric ozone to UTLS ozone and highlight its role for the radiative impact, or showing the impact on near-surface ozone, which is relevant for human health. Both aspects are mentioned here and there (near-surface more in the introduction, UTLS more in the conclusions), but I am missing the focus.

**Response (7):** We appreciate the reviewer's concern and are sorry for the confusion. The revised manuscript focuses exclusively on the contribution of stratospheric ozone to UTLS ozone and highlights its role in radiative impact. In our revised methodology, after detrending the ozone data over the study period (1962-2018), we found that the near-surface signal is weak and not significant; accordingly, we have removed the discussion of surface ozone impacts.

8. In the big picture, I am missing a comparison to different regions of similar latitudes than India: Are the stratosphere-troposphere exchange processes after SSWs similar here, or is the Indian region outstanding? If so, why?

**Response (8):** We thank the reviewer for this valuable suggestion. The present study is focused on ozone intrusion during SSWs over South Asia and its radiative forcing. While comparison with other regions at similar latitudes could yield useful insights, it is beyond the scope of the current analysis. We will, however, consider this aspect in future work.

**Specific points:**

9. L72: I would not call "wildfires" or "sea ice melt" tropospheric weather phenomena, but rather consequences of these.

**Response (9):** Thank you for this valuable suggestion. In the revised manuscript, we have removed the wording "tropospheric weather phenomena"(L63-66).

10. L83: The word "displaced" is used to explain the "displaced case", which is a circular explanation.

**Response (10):** Thanks for pointing this out. The term has been removed from the revised manuscript.

11. L84: Is the term "baby vortices" really the scientific term, or rather slang?

**Response (11):** Thanks for pointing this out. The term "baby vortices" has been removed from the revised manuscript.

12. L124: I think this is a typo: Typically ERA5 has 137 pressure levels, not just 37.

**Response (12):** We appreciate the reviewer's concern. The 137 levels refer to model/hybrid sigma-pressure levels from the surface up to ~0.01 hPa, while 37 levels refer to standard pressure levels (from the surface to 1 hPa), which are used in our current analyses. We have corrected the manuscript to explicitly state "37 standard pressure levels" (L125).

13. L143: It seems like the software has also an acronym (at least some characters in the name are capitalized). Please also give the common acronym.

**Response (13):** In the revised manuscript, we have included the common acronym, DISORT (Discrete Ordinate Radiative Transfer) code (L156).

14. L250: "Figure 2c-d shows clear evidence of a substantial ozone increase...": I do not agree with this statement. I think the authors cannot write about "clear evidence of a substantial ozone increase" in general. Actually, I think that only at 850 hPa one can see an increase over time. At

150 hPa, I would argue that one can see a decrease over time with an extreme outlier for 2018. Further, I think that this outlier should be discussed in detail. Further, the language should be more specific here and it should be mentioned that the discussed ozone increase is an increasFe over time over several years.

**Response (14):** In our revised methodology, after detrending the ozone data over the study period (1962–2018), we found that the near-surface signal is weak and not statistically significant. Accordingly, Fig. 2d (850 hPa) has been removed.

The reviewer may kindly note that Fig. 2c (in the old manuscript) does not represent a long-term increase over several years, but rather event-specific ozone enhancements linked to stratospheric intrusions during particular SSW years. However, this figure has been removed in the revised manuscript. Additionally, as suggested by the reviewer, the outlier behaviour of the 2018 event is discussed in detail in the revised manuscript (Section 3.1).

15. Figure 4: Why is 200 hPa chosen as a UTLS level, while in Fig.2, 150 hPa have been shown? Are there good reasons for that, or would it be possible to have the selection of pressure layers shown here to be consistent?

**Response (15):** Thanks for the valuable suggestion. To maintain consistency across the figures, we have chosen 200 hPa as a representative level for the upper troposphere in the revised manuscript.

16. L322: "Our analysis shows strong vertical coherence between 10 and 200 hPa levels": How can one observe this "strong vertical coherence" by comparing these plots? Please elaborate.

**Response (16):** We appreciate the reviewer's concern. In Figure 4, we observed that regions with low GPH at 10 hPa (Fig. 4a-e in the old manuscript) correspond to similar low GPH anomalies at 200 hPa (Fig. 4f-j in the old manuscript). To describe this feature, we used the term 'strong vertical coherence'. However, in the revised manuscript, this sentence is removed and the section has been modified as per our revised methodology.

17. L324: Please explain "wave-1" and "wave-2".

**Response (17):** 'Wave-1' and 'wave-2' refer to the zonal wavenumber-1 and wavenumber-2 components of planetary-scale disturbances in the stratosphere. In geopotential-height (GPH) anomaly maps (e.g., at 10 hPa; Fig. 4a-e in the old manuscript), wave-1 appears as a single ridge–trough dipole encircling the pole. Whereas wave-2 appears as two alternating ridge–trough pairs (e.g., Severe et al., 2013). However, in the revised manuscript, we have removed this term.

18. Figure 4 nicely showed the region of interest with a box. I suggest to repeat that box for the panels of Figure 5. Further, the orientation of the maps are different between these figures. Also

in panels a-b, it would be good to mark the boundaries of this box, since the text is mentioning that region.

**Response (18):** Figure 5 has been modified in the revised manuscript as per new methodology. However, as suggested by the reviwer, we showed the region of interest with a box and changed the orientation of the spatial maps (Fig 5a-b).

19. Section 3.3: These radiative forcings highlight again the outstanding role of the 2018 event, which is in focus of this paper, but this outstanding role is not sufficiently commented here.

**Response (19):** We thank the referee for pointing this out. In the revised manuscript, we have expanded the discussion of the 2018 event in detail (see section 3.1).

20. Figure 6: I am not sure about the relevance of this figure, since it basically shows 6 numbers, which are already given in the text.

**Response (20):** Thank you for pointing this out. Since Fig. 6 essentially repeated the values already presented in the text, we have removed this figure from the revised manuscript.

21. Line 414: I think the 290% given here are linked to the outlying event in 2018 and should not be given here as a general trend.

**Response (21):** As suggested by the reviewer, this sentence in the conclusion section has been modified in the revised manuscript (L420-422).

22. L439: I do not understand the final sentence: please rephrase.

**Response (22):** Since our focus is on the UTLS ozone in the revised manuscript, this sentence is removed.

23. L444: The given link is not accessible to me.

**Response (23):** The link has been updated in the revised manuscript.

24. Figure S1: Why is this figure given? It shows very, very similar results as Figure 1, just that the climatology is not subtracted from the values. Why do the authors want to show this figure in addition?

**Response (24):** Thank you for pointing this out. Since our present analysis is based on anomalies derived from the composites of all non-SSW years, we have removed Fig. S1 in the revised manuscript.

25. Caption Figure S1: Panel (b) is specifically mentioned, while panel (a) is not. I guess this was just missed to mention.

**Response (25):** In the revised manuscript, Figure S1 is removed.

Citation: https://doi.org/10.5194/egusphere-2025-1098-RC1

---

## Author Comment (AC3)

**Replies to the reviewer-III**

The study addresses sudden stratospheric warmings, especially split-downward propagating ones, and their impact on ozone increment over South Asia from the upper troposphere-lower stratosphere to the near-surface. Using the ERA 5 dataset from 1962 to 2018, the authors classified SSW events into three groups: split-dSSW, displace-dSSW, and nSSW. The paper showed a significant increase in ozone over the UTLS and near-surface around the SSW onset. The paper further focuses on the 2018 SSW event among 12 split-dSSWs to understand the dynamics, and concluded that Rossby wave breaking intensification relevant to SSW is the cause of the ozone increment. The authors additionally calculated the radiative forcing of such ozone increment.

Roy et al. covers relevant scientific questions within the scope of ACP, but needs major revision. To be specific, the dynamic linkage with SSW and ozone increment is weak and needs further explanation. Also, the significance test and the year 2018 as a representative case are not convincing enough. Lastly, figures and writing could be improved.

**Response:** We sincerely thank the reviewer for the meticulous evaluation, constructive comments, and valuable suggestions. In the revised manuscript, we have addressed the motivation for highlighting the 2018 SSW event and clarified the underlying mechanism. Our revised analysis reveals that, phase of the Quasi-Biennial Oscillation (QBO) plays a more critical role in shifting the subtropical jet position and associated Rossby wave breaking over the South Asian region during SSWs. Therefore, in the revised manuscript, we have reclassified the SSWs into two categories based on the prevailing QBO phase: SSWs coinciding with the westerly phase (WQBO-SSW) and those coinciding with the easterly phase (EQBO-SSW). Our results indicate that WQBO-SSW events are associated with a large positive ozone anomaly over the South Asian region compared to EQBO-SSW events. We have also incorporated all recommendations and improved the overall structure. All changes are marked in the track-change version of the manuscript, with line numbers referenced in the responses.

**Major comments**

1. How can ozone response prior to and close to the SSW event date be caused by SSW? It requires some time for the response to propagate to the UTLS midlatitudes and low latitudes.

**Response (1):** We agree with the reviewer. However, we would like to emphasize that, unlike high latitudes, where SSWs exert a direct downward influence on the troposphere, our analysis indicates that the low-latitude responses (such as over South Asia) are mediated primarily by Rossby-wave dynamics. Specifically, by RWB and PV streamer intrusions along the subtropical waveguide. The location of these RWBs is modulated by the positioning of the subtropical jet during SSWs. It is well established that major SSWs are preceded by enhanced mid-latitude

planetary and synoptic wave driving (e.g., Baldwin et al., 2021). Whether and how that wave activity projects into South Asia depends on the background flow set by the QBO. During the westerly QBO, the associated secondary circulation warms the equatorial lower stratosphere and cools the subtropics, sharpening and shifting the UTLS meridional temperature gradient equatorward (e.g., Hitchman et al., 2021). By thermal-wind balance, this strengthens upper-tropospheric westerlies on the equatorward flank and displaces the subtropical jet equatorward over the South Asian longitudes, favouring subtropical wave guidance, RWB, and PV-streamer intrusions. We now detail this mechanism in Section 3.1.1.

(a) Baldwin, M. P., Domeisen, D. I. V., Hegglin, M. I., Garny, H., Garfinkel, C. I., Langematz, U., Charlton-Perez, A. J., Butchart, N., Gerber, E. P., Birner, T., Butler, A. H., Ayarzagüena, B., and Pedatella, N. M.: Sudden Stratospheric Warmings, Reviews of Geophysics, 59, https://doi.org/10.1029/2020rg000708, 2021.

(b) Hitchman, M. H., Tegtmeier, S., Yoden, S., Haynes, P. H., and Kumar, V.: An Observational History of the Direct Influence of the Stratospheric Quasi-biennial Oscillation on the Tropical and Subtropical Upper Troposphere and Lower Stratosphere, Journal of the Meteorological Society of Japan. Ser. II, 99, 239–267, https://doi.org/10.2151/jmsj.2021-012, 2021.

2. The novelty and motivation of the study should be emphasized more. Although not focused on South Asia, previous studies like William et al. (2024) and Lee et al. (2025) have already calculated the radiative impact and global ozone anomaly due to SSW. I think the authors showed an interesting result that only split-dSSW has a substantial impact, while other types of SSW do not. I recommend emphasizing this more and providing additional explanations for why.

**Response (2):** Thank you for the helpful suggestion. Our revised analysis highlights that SSWs occurring during the westerly phase of the QBO (WQBO-SSW) are associated with a more pronounced equatorward shift of the subtropical jet, enhanced Rossby wave breaking, and deeper ozone intrusions over South Asia compared to SSWs coinciding with the easterly phase (EQBO-SSW). The detailed mechanisms underlying these features are discussed in the revised manuscript (section 3.2).

3. When introducing the previous studies and comparing, please match the units and consider the time scale. For example, % and ppb units are hard to compare and confuse their importance. Also, stratospheric intrusion (hourly to daily) should be carefully compared to the stratosphere-troposphere exchange (often in seasonal, interannual, and even climatology) or trend due to anthropogenic emissions.

**Response (3):** We appreciate the reviewer's suggestion. In the revised manuscript, we harmonise units and timescales when introducing prior work and making comparisons. We also have modified the sentences where stratospheric intrusions are compared with STE and the trend due to anthropogenic emissions.

4. Is the Student t-test appropriate in this sample size? I do not understand how a significance test is being conducted on a single case (year 2018). Also, 12 cases of split-dSSWs are pretty small. I would recommend adding more detail on how the significance test is done and improving the statistical robustness of the result.

**Response (4):** We apologise for not describing our statistical testing in sufficient detail. We have now added the details on how the statistical tests are done in the Methods section of the revised manuscript (L134-141).

Given the small sample size for and the likelihood of non-normal distributions, we replaced the initial Student's *t*-test with the Monte Carlo bootstrap and the Wilcoxon signed-rank test in the revised manuscript. "For the Monte Carlo, we built a calendar-matched null by resampling days from non-SSW years within the same day-of-year window. We then use a bias-corrected and accelerated (BCa) bootstrap with 20,000 resamples to form 95% confidence intervals. For 2018, we checked whether the observed value lay outside the BCa interval of the background ensemble. For the composite, we tested whether the mean anomaly differed from zero. Next, we applied an exact Wilcoxon signed-rank test to the same data. A grid point is called significant only when both tests agree at 95% significance." This approach offers three advantages: (a) distribution-free inference suitable for small samples, (b) improved coverage from BCa intervals that correct bias and skewness, and (c) robustness of the Wilcoxon test to outliers and non-Gaussianity (Efron, 1987; Efron & Tibshirani, 1993; Davison & Hinkley, 1997; Wilcoxon, 1945)."

   (a) Efron, B.: Better Bootstrap Confidence Intervals, Journal of the American Statistical Association, 82, 171–185, https://doi.org/10.1080/01621459.1987.10478410, 1987.
   (b) Efron, B. and Tibshirani, R. J.: An Introduction to the Bootstrap, chapman hall crc, https://doi.org/10.1201/9780429246593, 1994..
   (c) Davison, A. C. and Hinkley, D. V.: Bootstrap Methods and their Application, cambridge university, https://doi.org/10.1017/cbo9780511802843, 1997.
   (d) Wilcoxon, F.: Individual Comparisons by Ranking Methods, Biometrics Bulletin, 1, 80, https://doi.org/10.2307/3001968, 1945.

5. Can the year 2018 represent the split-dSSWs? Figures 2 and 3 compare the year 2018 with the other split-dSSWs. However, the year 2018 seems to be anomalously strong, and the patterns look quite different. If there is a reason to choose the year 2018, please explain it. Also, as stated in the paper, due to anthropogenic emissions and global warming, tropospheric ozone has a positive tendency. If this is not detrended, could there be a bias in the impact of SSW in the year 2018?

**Response (5):** We appreciate the reviewer's concern. We have examined all major SSWs from 1962 to 2018. Our analysis revealed a relatively more equatorward shift of the subtropical jet over South Asia and associated large ozone intrusion during the 2018 SSW compared to other

SSW years. This motivated us to report the detailed mechanism of the 2018 SSW as a case study. We have included this motivation in the introduction section of the revised manuscript (L97-100). This equatorward shift facilitates the eastward-propagating synoptic-scale Rossby waves to move further equatorward, favouring RWB and PV-stremer activity along with more ozone intrusion (Homeyer & Bowman, 2013; Albers et al., 2016). This unique dynamical setup explains why the 2018 event stands out as an outlier in the UTLS ozone response. In the revised manuscript, we have discussed this aspect in detail (see section 3.1.1).

To the reviewer's comment on "trend", to mitigate potential bias from long-term tropospheric ozone trends due to anthropogenic emissions and global warming, we detrended the ozone data over 1962–2018 in the revised methodology. The enhancement in 2018 is therefore interpreted as SSW-related dynamical transport rather than an artefact of the trend.

6. Authors emphasized the impact on the near-surface ozone level from SSW, but detailed explanations are missing. How can the signal in the UTLS propagate toward the near-surface? The propagation of geopotential height in the figure is difficult to recognize. Also, tropospheric ozone has multiple sources, such as local emissions and tropospheric long-range transport. What is the contribution of each driver to the ozone increment, and is the stratospheric intrusion truly the dominant driver?

**Response (6):** We appreciate the reviewer's concern. In our revised methodology, after detrending the ozone data over the study period (1962-2018), we found that the near-surface signal is weak and not significant. Accordingly, we have removed the discussion of surface impacts. The revised manuscript now focuses exclusively on the UTLS region, where the stratospheric intrusion signal is robust and more directly attributable to SSW-related dynamics.

7. It seems the Rossby wave breaking is the suggested mechanism for the ozone intrusion. However, it is hard to see that RWB has been amplified or more frequent due to the split-dSSWs. It would be helpful to explicitly show that the frequency or amplitude of the RWB, whichever matters, differs by type of SSW. The current qualitative discussion with Fig. 5 is not convincing enough.

**Response (7):** We appreciate the reviewer's concern. However, the reviewer may kindly note that the categorization of SSW has been changed in the revised manuscript. We now consider all the major SSWs during the study period and categorize them based on the QBO phase. This has been mentioned in the methodology of the revised manuscript. Further, in the revised manuscript, we present the longitude-pressure composite of PV (2 PVU contour) averaged over the South Asian region for both the western and easterly phases of the QBO (Fig. 6). During the westerly phase of the QBO, the 2 PVU composite extend below 300 hPa into the upper troposphere. This behaviour is consistent with stronger and more frequent RWB in WQBO compared to EQBO, facilitating deeper PV-streamer penetration over the region.

8. Is the radiative forcing due to SSW significant over this region? If so, further information to compare and show the importance could be helpful. Also, what is the temperature change due to this radiative forcing?

**Response (8):** The radiative forcing due to SSW is statistically significant ($p < 0.05$) and has a warming effect over South Asia. All RFs are positive and exceed the uncertainty range ($\pm 1$ standard deviation). This kernel-diagnosed RF reflects the event-scale radiative response to SSW-related ozone/temperature anomalies and, by design, is not directly comparable to published effective radiative forcing from long-term changes in ozone or anthropogenic influence. Hence, we opt to refrain from a comparison and instead report the magnitude ($W.m^{-2}$)

To the reviewer's comment on "temperature change due to this radiative forcing…", in this study, our focus has been on examining ozone intrusions into the UTLS during SSWs and their associated radiative impacts. Hence, we did not explicitly quantify the resulting temperature changes due to radiative forcing. We acknowledge that establishing a link between ozone-induced radiative forcing and UTLS temperature response is an important avenue for future investigation.

Lee, J., Butler, A. H., Albers, J. R., Wu, Y., & Lee, S. H. (2025). Impact of sudden stratospheric warmings on the stratosphere-to-troposphere transport of ozone. Geophysical Research Letters, 52(2), e2024GL112588.

**Minor comments**

9. L47: What does this 'increase' mean? A trend or variability?

**Response (9):** Thanks for pointing this out. We have removed this line from the revised manuscript.

10. L48-50: Add reference to show the contribution of the stratosphere to the South Asian ozone

**Response:** Thanks for pointing this out. In the revised manuscript, references are added (L45-46).

11. L98: Isn't Lu et al. (2023) only focused on the stratospheric ozone variance?

**Response (11):** We acknowledge the reviewer's concern. After revisiting Lu et al. (2023), we recognize that their work is primarily focused on stratospheric ozone variability. Accordingly, we have removed the reference.

12. L99-100: Willims et al. (2024) only focused on the PJO type of SSW when leading to that conclusion. Please clearly state it.

**Response (12):** Thanks for pointing this out. We have now revised the sentence to accurately reflect the scope of the original study (L79-81).

13. L111: Maybe split-dSSWs? Not all the downward-propagating SSWs?

**Response (13):** Thanks for pointing this out. In the revised version, we investigated the impact of all the SSW events from 1962 to 2018 on ozone variability in the UTLS over the South Asia region (L92-94).

14. L127-129: What does the corresponding daily mean of all the non-SSW days mean? Is every day except SSW onset days considered?

**Response (14):** We appreciate the reviewer's concern. The daily climatology is constructed using data from non-SSW years during the extended winter season (November–May). Anomalies were computed by subtracting this daily climatology from the respective calendar days of SSW years. We have clarified this point in the revised manuscript (L130-132).

15. L152: UTLS definition should be included.

**Response (15):** Thanks for pointing this out. In this study, we consider the upper troposphere and lower stratosphere (UTLS) region to span the pressure levels from 300 hPa to 50 hPa, consistent with previous literature (e.g., Chavan et al., 2021). We have added this in the revised manuscript (L93).

   (a) Chavan, P., Fadnavis, S., Griessbach, S., Chakroborty, T., Sioris, C. E., and Müller, R.: The outflow of Asian biomass burning carbonaceous aerosol into the upper troposphere and lower stratosphere in spring: radiative effects seen in a global model, Atmospheric Chemistry and Physics, 21, 14371–14384, https://doi.org/10.5194/acp-21-14371-2021, 2021.

16. L159-163: These two sentences seem to be repeated.

**Response (16):** Thanks for pointing this out. We have removed the sentence from the revised manuscript to avoid repetition.

17. L193-195: Hall et al. (2021) used ERA-Interim and ERA 40 for the classification. Is ERA5 consistent with these two datasets?

**Response (17):** In most cases, ERA5 shows higher accuracy than that of ERA-Interim and ERA-40 (Hersbach et al., 2020, S-RIP Final Report, 2024).

18. L210-212: Why is ± 30 days selected when the previous sentence mentioned up to 60 days after SSW onset?

**Response (18):** We appreciate the reviewer's concern. In the revised manuscript, we have focused on the ±30-day window, as this period best captures the dominant signal over the study region (L126–127). Since the most pronounced ozone enhancement occurs within ±6 days of the SSW onset, all subsequent analyses in this study are conducted for this period (L202-203).

19. L220: I think this is a typo. Maybe section 3.2, not 2.2?

**Response (19):** We thank the reviewer for bringing this to our attention. Yes, this was a typographical error. The sentence has been removed in the revised manuscript.

20. L237-239: It is hard to see the downward propagation of GPH in the figure

**Response (20):** Thanks for pointing this out. In the revised manuscript, we have removed this term.

21. L239-241: I can't see the lowering of the 380K. Also, why is the 380K isoline being used here?

**Response (21):** We agree that the lowering of the 380 K potential temperature isoline is not clearly visible in the figure. Accordingly, we have removed this sentence. We have used the 380 K potential temperature isoline as a proxy for the tropopause in tropical and subtropical regions (Fueglistaler et al., 2009). Examining the 380K contour alongside ozone and GPH anomalies provides a meaningful way to identify stratosphere-to-troposphere exchanges during SSW events.

    (a) Fueglistaler, S., Mote, P. W., Folkins, I., Fu, Q., Dessler, A. E., and Dunkerton, T. J.: Tropical tropopause layer, Reviews of Geophysics, 47, https://doi.org/10.1029/2008rg000267, 2009.

22. L252-255: What leads to the two different peaks in 2018?

**Response (22):** Thanks for pointing this out. In our revised methodology, after detrending the ozone data over the study period (1962–2018), we found that the near-surface signal is weak and not statistically significant. Accordingly, this discussion and the related figure (Fig S6) are removed from the revised manuscript.

23. L282-283: I don't think this has been discussed later in this section.

**Response (23):** Thank you for bringing this to our attention. This aspect is discussed in Section 3.1.1, not in the current section. We have corrected this in the revised manuscript accordingly (L207-208).

24. L283-286: Where is the anomalous lowering of the tropopause? Figure 3 doesn't show the anomalous tropopause.

**Response (24):** We thank the reviewer for pointing this out. The term "anomalous lowering of the tropopause" has been removed, and the sentence is modified in the revised manuscript (L208-210).

25. L323-326: It is hard to see the wave-1 and wave-2 pattern in Fig. 4. Also, please explain the connection with vertically coherent waves at 10 hPa and 200 hPa to the subtropical jet perturbation. Currently, there is no explanation why the first sentence leads to the conclusion in the next one.

**Response (25):** We appreciate the reviewer's concern. In Figure 4 (in the old manuscript), GPH anomaly maps allow identification of planetary wave structures: wave-1 is characterised by a single ridge–trough dipole encircling the pole, while wave-2 manifests as two alternating ridge–trough pairs (e.g., Seviour et al., 2013). However, in the revised manuscript, we have removed this term.

In Figure 4 (in the old manuscript), we observed that regions with low GPH at 10 hPa (Fig. 4a-e in the old manuscript) correspond to similar low GPH anomalies at 200 hPa (Fig. 4f-j in the old manuscript). To describe this feature, we used the term 'strong vertical coherence'. However, in the revised manuscript, this sentence is removed, and the section has been modified as per our revised methodology.

26. L338-340: Which pattern is the persistent GPH anomaly over South Asia? A more detailed description would be helpful.

**Response (26):** We acknowledge the reviewer's concern. The persistent pattern corresponds to low GPH over South Asia. A more detailed description of this feature has been added in the revised manuscript (L231-233).

27. Figure 4: It is hard to follow the explanation without the latitude grid line

**Response (27):** Thanks for this suggestion. The grid line is added in Figure 4 of the old manuscript (Fig. 2 in the revised manuscript).

28. L368-369: Please explain in more detail about the synoptic wave pattern that only exists in Fig. 5c. It is hard to see in this figure. Also, the figure is rotated from Fig. 4, which is confusing.

**Response (28):** We acknowledge the reviewer's concern and apologize for the confusion. We ensured that in the revised manuscript, the orientation in Figure 5 (Fig. 5b-d in the revised manuscript) is consistent with Figure 4, and a detailed explanation is added (L350-351).

29. L370-371: What does it mean to enhance RWB? Is it amplitude-wise or frequency-wise?

**Response (29):** In our analysis, "enhanced RWB" refers to both its frequency and amplitude. In the revised manuscript, we present the longitude-pressure composite of PV (2 PVU contour) averaged over the South Asian region for both the western and easterly phases of the QBO (Fig. 6). During the westerly phase of the QBO, the 2 PVU composite extend below 300 hPa into the upper troposphere. This behaviour is consistent with stronger and more frequent RWB in WQBO compared to EQBO, facilitating deeper PV-streamer penetration over the region.

30. L432-435: A reference is needed for the future projection.

**Response (30):** Thanks for pointing this out. Reference is added in the revised manuscript (L431).

(a) Kim, J., Park, H.-S., Son, S.-W., and Gerber, E. P.: Defining Sudden Stratospheric Warming in Climate Models: Accounting for Biases in Model Climatologies, Journal of Climate, 30, 5529–5546, https://doi.org/10.1175/jcli-d-16-0465.1, 2017.

31. L435-436: A reference is needed for the high-top model performance expectation.

**Response (31):** Thanks for pointing this out. Reference is added in the revised manuscript (L436-438).

(a) Scaife, A. A., Charlton-Perez, A. J., Son, S.-W., Hardiman, S. C., Polvani, L., Lim, E.-P., Haynes, P., Baldwin, M. P., Shepherd, T. G., Perlwitz, J., Richter, J. H., Noguchi, S., Thompson, D. W. J., Karpechko, A. Y., Butler, A. H., Scinocca, J., Sigmond, M., Domeisen, D. ShiI. V., and Garfinkel, C. I.: Long-range prediction and the stratosphere, Atmospheric Chemistry and Physics, 22, 2601–2623, https://doi.org/10.5194/acp-22-2601-2022, 2022.